# PaRa: Personalizing Text-to-Image Diffusion via Parameter Rank Reduction

**Shangyu Chen & Zizheng Pan & Jianfei Cai & Dinh Phung**
Monash University
Melbourne, Australia
{shangyu.chen,zizheng.pan,jianfei.cai,dinh.phung}@monash.edu

## Abstract

Personalizing a large-scale pretrained Text-to-Image (T2I) diffusion model is challenging as it typically struggles to make an appropriate trade-off between its training data distribution and the target distribution, *i.e.*, learning a novel concept with only a few target images to achieve personalization (aligning with the personalized target) while preserving text editability (aligning with diverse text prompts). In this paper, we propose **PaRa**, an effective and efficient **Pa**rameter **Ra**nk Reduction approach for T2I model personalization by explicitly controlling the rank of the diffusion model parameters to restrict its initial diverse generation space into a small and well-balanced target space. Our design is motivated by the fact that taming a T2I model toward a novel concept such as a specific art style implies a small generation space. To this end, by reducing the rank of model parameters during finetuning, we can effectively constrain the space of the denoising sampling trajectories towards the target. With comprehensive experiments, we show that PaRa achieves great advantages over existing finetuning approaches on single/multi-subject generation as well as single-image editing. Notably, compared to the prevailing fine-tuning technique LoRA, PaRa achieves better parameter efficiency ($2\times$ fewer learnable parameters) and much better target image alignment.

## 1 Introduction

Recent text-to-image (T2I) diffusion models (Rombach et al., 2022; Podell et al., 2023; OpenAI, 2023; MidJourney) have achieved unprecedented success. However, despite being trained on large-scale datasets, most T2I models struggled to generate novel concepts as they were limited within their training data distribution. For example, pretrained Stable Diffusion (SD) models (Rombach et al., 2022) cannot generate unseen objects like a novel anime character. Thus, it has drawn increasing attention in the community to teach off-the-shelf T2I models with a few target images to learn a novel concept (*e.g.*, a specific personal pet) for aligning with user preferences, which is known as T2I model personalization.

Much effort has been made in this direction. Some works (Jia et al., 2023; Wei et al., 2023) seek for a general encoder to learn a novel concept without test-time finetuning. However, training such an encoder usually requires building a large collection of text-image pairs and expensive computing resources, *e.g.*, 1.4 million pairs for the category of person in InstantBooth (Shi et al., 2023) and 128 TPUv4 chips in SuTI (Chen et al., 2024). Another prevalent line of work directly finetunes T2I models based on the target images, which can be roughly categorized into using text embedding (Gal et al., 2022), cross-attention layers (Kumari et al., 2023), full model finetuning (Ruiz et al., 2023), low-rank update (Hu et al., 2021; cloneofsimo, 2022) or adjusting singular values of model parameters (Han et al., 2023). However, all these fine-tuning methods directly change the initial full generation space, which naturally results in a trade-off between generation diversity and the alignment with the target concept. Consequently, they usually suffer from either poor alignment on the new concept (Gal et al., 2022), or overfitting the few target text-image pairs (Ruiz et al., 2023).

In this paper, we introduce PaRa, a new parameter-efficient framework for T2I model personalization via **Pa**rameter **Ra**nk Reduction. Our motivation comes from two folds: First, diffusion models are trained to capture their training data distribution, indicating a diverse image generation

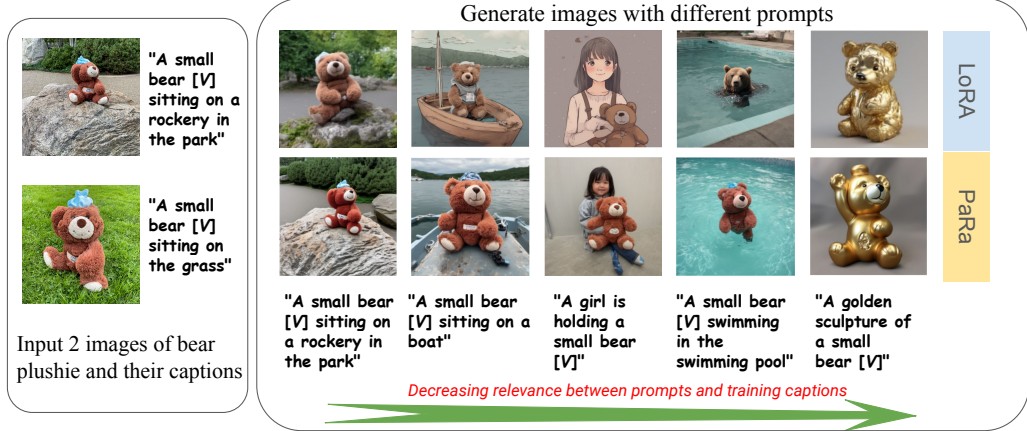

Figure 1: Comparison between LoRA (cloneofsimo, 2022) and our propsoed PaRa on T2I personalization for learning a new concept of bear, *i.e.*, "*[V]*". For a fair comparison, we set the rank as 4 and adopt the same latent noise for both methods. LoRA scale is set to 1.0.

space due to large-scale pretraining. However, taming a T2I model for a novel concept may instead suggest a small restricted generation space. For example, finetuning SDs into a specific anime style like Ghibli Studio[1] does not requires strong photorealistic knowledge learned from LAION-5B dataset (Schuhmann et al., 2022). Second, in a broad literature, the rank of matrix has been shown to be effective to control the degree of certain restrictions such as in image compression (Andrews & Patterson, 1976), network pruning (Idelbayev & Carreira-Perpiñán, 2020) and parameter-efficient finetuning (Hu et al., 2021).

Based on the discussions above, our key idea is to explicitly control the rank of a layer output in a diffusion model during the denoising sampling, thus effectively steering the image generation into a well-aligned and restricted space for learning a new concept. In practice, this is achieved by introducing a low-rank learnable parameter $B \in \mathbb{R}^{d \times r}$ for a pretrained linear projection $W_0 \in \mathbb{R}^{d \times k}$, where $r$ is the reduced rank and $d$ is the number of hidden dimensions of this layer. Next, with a simple QR decomposition of $B = QR$, we can obtain a set of orthonormal bases $Q$ which can be adopted to reduce the rank of the initial layer outputs by $Wx - QQ^T Wx$, where $x$ denotes the layer input. We theoretically prove this result in Appendix .1. Note that this is fundamentally different from LoRA (cloneofsimo, 2022; Hu et al., 2021), which adds low-rank updates to a layer output to achieve parameter-efficienct finetuning.

With extensive experiments, we demonstrate several advantages of PaRa. First, compared to the typical LoRA fine-tuning (cloneofsimo, 2022), PaRa can generate images far beyond its initial training prompts while maintaining well the consistency of the specific target concept, as shown in Figure 1. Furthermore, PaRa only requires half of the storage cost of LoRA since it only needs one matrix $B$ for each layer. Second, benefiting from the explicit rank control, we further propose an approach to combine multiple individually fine-tuned PaRa weights, which enables the blending of multiple personalized concepts for multi-subject T2I generation. without the need for additional training with augmented multi-subject images (Han et al., 2023). Third, PaRa is fully compatible with LoRA weights (Hu et al., 2021), supporting the combination of PaRa with a wide range of existing LoRA finetuned SD models in the public model zoo. Finally, thanks to the restricted small generation space, single image editing under PaRa is quite stable as the initial noise has a negligible effect on the final image generation. For example, as demonstrated in the experimental section 4.3, under the same prompts, PaRa achieves a significantly higher average SSIM, compared to the baseline.

The major contributions of this work can be summarized as follows.

- We propose Parameter Rank Reduction (PaRa), a new framework for personalizing T2I diffusion models by explicitly reducing the rank of the diffusion model parameters.

---

[1]https://huggingface.co/nitrosocke/Ghibli-Diffusion

- We propose a simple yet effective approach to combine multiple individually fine-tuned PaRa models, enabling the blending of multiple personalized concepts. Our model also facilitates single-image editing without the need to do the noise inversion (Song et al., 2020a; Mokady et al., 2023) in the diffusion process.
- We conduct comprehensive experiments to show that PaRa achieves state-of-the-art performance in personalized single/multi-subject generation.

## 2 RELATED WORK

### 2.1 TEXT-TO-IMAGE DIFFUSION MODELS

Generative models exemplified by Diffusion Models(Chang et al., 2023; Gu et al., 2022; Ho et al., 2020; Nichol & Dhariwal, 2021; Song et al., 2020a; Sohl-Dickstein et al., 2015; Song et al., 2020b) have achieved significant advancements in the past few years. Ho et al. (2020) proposed the Denoising Diffusion Probabilistic Models (DDPM), which enable diffusion models to achieve excellent performance in image generation tasks. Song et al. (2020a) further improved this approach by combining score function with diffusion probabilistic models, significantly enhancing the efficiency of image generation. DALL-E 2 (Ramesh et al., 2022), Imagen (Saharia et al., 2022), and Stable Diffusion (Rombach et al., 2022) iteratively denoise data within a latent space, trained on large image-text datasets, significantly enhancing the practicality and effectiveness of diffusion models.

Among the existing T2I diffusion models, Stable Diffusion (SD) (Rombach et al., 2022) is the most widely used one, which is a variant of Latent Diffusion Models (LDMs) (Rombach et al., 2022). LDMs encode input images $x$ into a latent code $z$ with encoder $\mathcal{E}$, and then execute the denoising process on $z$. The training objective is to minimize

$$\mathbb{E}_{\mathcal{E}(x),y,\epsilon\sim\mathcal{N}(0,1),t}\left[\|\epsilon - \epsilon_\theta\left(z_t, t, \tau(y)\right)\|_2^2\right] \tag{1}$$

where $y$ is the input condition such as text or semantic maps, $\tau$ is a domain-specific encoder of $y$, $t$ is the sampling step, $\epsilon_\theta\left(z_t, t, \tau(y)\right)$ is the conditional denoising model with parameters $\theta$. The denoising process is modeled as a reverse process of a fixed Markov Chain of length $T$, and $t$ is uniformly sampled from $\{1, \ldots, T\}$.

SD uses a frozen CLIP text encoder (i.e. $\tau$) to encode prompt words (i.e. $y$) and a UNet Model (i.e. $\epsilon_\theta$) for the denoising process (Rombach et al., 2022). Our experiments primarily rely on Stable Diffusion. The forward linear units in its UNet Model are fundamental to our proposed rank reduction.

### 2.2 FINE-TUNING GENERATIVE MODELS FOR PERSONALIZATION

Customizing and personalizing pre-trained text-to-image diffusion models has garnered significant research interest recently. Many methods have been proposed including enriching text embeddings (Gal et al., 2022), fine-tuning the UNet (Ruiz et al., 2023; Kumari et al., 2023; Hu et al., 2021; Gandikota et al., 2023; Han et al., 2023; cloneofsimo, 2022; Qiu et al., 2023; Yeh et al., 2023; Marjit et al., 2024), and providing adapters (Mou et al., 2024; Zhang et al., 2023a) to control the generated outcomes, as well as some training-free approaches (Chen et al., 2024; Gal et al., 2023; Jia et al., 2023; Shi et al., 2023; Wei et al., 2023).

Among the fine-tuning based personalized T2I methods, one common and effective idea is via matrix decomposition and adjustment of its components, which have been introduced in early GAN-based generative models (Zhu et al., 2021; Feng et al., 2022). The widely-used LoRA (Hu et al., 2021; cloneofsimo, 2022; Gandikota et al., 2023) in diffusion models also follows the idea of matrix decomposition. Another method SVDiff (Han et al., 2023) uses SVD (singular value decomposition) to decompose matrices. However, these methods only focus on the scale of the obtained vector components. For example, LoRA (cloneofsimo, 2022; Hu et al., 2021) adds the scale $\alpha$ to the original formula $W_0 + BA$ as

$$W_0 + \alpha\Delta W = W_0 + \alpha BA, \text{where } W_0 \in \mathbb{R}^{d\times k}, B \in \mathbb{R}^{d\times r}, A \in \mathbb{R}^{r\times k} \tag{2}$$

where $W_0$ is matrix of a pretrained SD model. Note that since the column vectors of $\Delta W$ are not normalized, $B$ and $A$ inherently include the learning of scales for the components. Similarly,

SVDiff (Han et al., 2023) optimizes the scales of the diagonal matrix obtained from the SVD decomposition of $W_0$.

Our PaRa method, in contrast, chooses to directly eliminate certain components during personalization rather than adjusting their scales. Because neural networks often tend to overfit, eliminating some components can be more stable and robust than adjusting their scales (Liu et al., 2015; Guo et al., 2016; Han et al., 2015). Furthermore, eliminating components is an idempotent operation (meaning it can be applied multiple times without changing the result to go beyond the initial application), making model mixing or combination more stable, compared to scale adjustments.

### 2.3 PERSONALIZATION-BASED IMAGE EDITING IN DIFFUSION MODELS

Existing fine-tuning models (Ruiz et al., 2023; Kumari et al., 2023; Hu et al., 2021; Gandikota et al., 2023; Han et al., 2023; Zhang et al., 2023b) that attempt to perform image editing directly by "one-shot training and adjusting the text of the original training image" are prone to overfitting to the single training image. Even if they can avoid overfitting, the diversity of images generated from the same text and Gaussian noise makes it nearly impossible to reproduce the exact training images during generation. Therefore, personalized diffusion models require the inversion process (Song et al., 2020a; Mokady et al., 2023) to lock in the noise during image editing. SVDiff (Han et al., 2023) demonstrates that it can treat the inversion process as an optional component. However, omitting the inversion process still significantly impacts its faithfulness to the target image. In contrast, our model PaRa can maintain editability after only one-shot learning, and eliminate the need for the inversion process to achieve image editing.

## 3 METHOD

This section introduces the details of our PaRa model, including three sections. Firstly, we explain the fundamental principles and implementation of PaRa, including its formulation, training process, and application to convolutional layers. Secondly, we discuss how to combine two trained PaRa models, what the relationship is between PaRa and LoRA, and how PaRa can be effectively used in conjunction with pre-trained LoRA models. Lastly, we delve into the application of PaRa in single-image editing.

### 3.1 PARAMETER RANK REDUCTION

In this section, we describe our approach PaRa with a simple demonstration based on a linear projection. Let $W_0 \in \mathbb{R}^{d \times k}$ be the weight matrix of one linear projection in a diffusion model, where $d$ and $k$ denote the number of input and output hidden dimensions, respectively. Given the input of this layer $x$, the output can be written as $h = W_0 x$. Note that we omit the bias term for simplicity. To reduce the output space (*i.e.*, column space) of $W_0$, we introduce a learnable parameter $B \in \mathbb{R}^{d \times r}$, where $r$ is the hyperparameter that controls the matrix rank. We first decompose $B$ using QR decomposition $B = QR$, where $Q$ is an orthogonal matrix and $R$ is the corresponding upper triangular matrix. Based on $Q$, we then compute $W_{reduce}$ as

$$W_{reduce} = W_0 - QQ^T W_0. \tag{3}$$

Next, the rank-reduced layer outputs can be formulated as

$$h = W_{reduce} x = W_0 x - QQ^T W_0 x. \tag{4}$$

With a theoretical proof provided in Appendix.1, Eq. 4 ensures that the column space of $W_{reduce}$ is a subset of the column space of $W_0$, effectively reducing the dimension of the output while maintaining the key features learned by the model.

In other words, given $W_0 = [\vec{w_1} \ \vec{w_2} \ ... \ \vec{w_k},]_{d \times k}$, where $\vec{w_i}$ denotes the column vector of $W_0$, we define the column space of $W_0$ as $S_0 = Span\{\vec{w_1}, \vec{w_2}, ... \vec{w_k}\}$. By adjusting $W_0$ to $W_{reduce}$, we ensure that the column space of $W_{reduce}$ is a subset of the column space of $W_0$. In practice, we initialize $B$ to zero and finetune it with a few text-image pairs as the common practice in (Ruiz et al., 2023). Our goal is to evolve $B$ into a set of orthonormal bases, and then adjust $W_0$ by subtracting its components on these bases. This process ensures the column space is effectively reduced.

Moreover, note that $QQ^T W_0$ now represents the components of $W_0$ projected onto the column space of $B$, capturing the influence of the orthonormal basis derived from $B$ on $W_0$. When $B$ has linearly independent columns, the reduced dimension will be the same as $r$, i.e. the column number of $B$. Since $r$ is typically set to a small number ( *e.g.*, 2 or 4, compared to hundreds/thousands of hidden dimensions in a layer output), $B$ is likely to have linearly independent columns after training.

**Remark on convolutional kernels.** For the weights of convolutional layers, we need to employ a reshaping method similar to FSGAN (Robb et al., 2020) before reducing the rank. Specifically, we reshape the convolution kernel weight $W_{0\_conv} \in \mathbb{R}^{c_{out} \times c_{in} \times h \times w}$ to the matrix as a second-order tensor $W_0 \in \mathbb{R}^{c_{out} \times (c_{in} \times h \times w)}$. After this, we can proceed with the steps of PaRa, setting $B \in \mathbb{R}^{c_{out} \times (c_{in} \times h \times w)}$, $B = QR$, calculate and reshape $QQ^T W_0$ from $c_{out} \times (c_{in} \times h \times w)$ to $c_{out} \times c_{in} \times h \times w$ as $\Delta W$. Finally, the rank-reduced kernel weight becomes

$$W_{reduce\_conv} = W_{0\_conv} - \Delta W, \tag{5}$$

where $\Delta W \in \mathbb{R}^{c_{out} \times c_{in} \times h \times w}$.

## 3.2 COMBINING PARA

Our framework also supports the combinations of different personalized PaRa models as well as combining with LoRA-based personalized T2I models. Let's first consider two individually trained PaRa models

$$W_1 = W_0 - Q_1 Q_1^T W_0, W_2 = W_0 - Q_2 Q_2^T W_0, \tag{6}$$

where $Q1 = [\vec{q_{11}} \ \vec{q_{12}} \ ...\vec{q_{1r_1}}]_{d \times r_1}$ and $Q2 = [\vec{q_{21}} \ \vec{q_{22}} \ ...\vec{q_{2r_2}}]_{d \times r_2}$. We can merge $Q_1$ and $Q_2$ into $Qm = [\vec{q_{11}} \ \vec{q_{12}} \ ...\vec{q_{1r_1}} \ \vec{q_{21}} \ \vec{q_{22}} \ ...\vec{q_{2r_2}}]_{d \times (r_1 + r_2)}$ and reduce to a new orthonormal matrix $Q'_m$ by QR decomposition $Q_m = Q'_m R'_m$. Then the combined PaRa model has new weights $W_m = W_0 - Q'_m Q'^T_m W_0$. In practice, a more convenient approach is via sequential addition to a diffusion model, where the first PaRa diffusion model is used as the new base diffusion model for the second one. This can be expressed as:

$$W_m = W_1 - Q_2 Q_2^T W_1 = (W_0 - Q_1 Q_1^T W_0) - Q_2 Q_2^T (W_0 - Q_1 Q_1^T W_0). \tag{7}$$

We provide a proof in the Appendix A that this is equivalent to $W_m = W_0 - Q'_m Q'^T_m W_0$.

**Rank boundary.** For a matrix $W_0$ with rank $k$, if the reduced rank $r$ is too large compared to $k$, it can cause the original model to collapse and fail to generate images properly. Especially in the PaRa combination, we are uncertain about the exact value of the combined rank $r_{combine}$, for which we only know that it is greater than $r_1$ and $r_2$, but less than $r_1 + r_2$. Thus, there is a possibility that $r_{combine}$ could be large enough to cause the original model to collapse. Therefore, for an PaRa, even though we set the same $r$ for the entire pre-trained model, for each layer's weight $W_0$ in the pre-trained diffusion model, we impose an upper limit on the rank $r$ that can be reduced, denoted as $r_{adjust} \leq \gamma rank(W_0)$, where $\gamma$ is a factor less than 1. For each $W_0$, the reduced rank is

$$r_{adjust} = \begin{cases} r & \text{if } r \leq \gamma rank(W_0) \\ \lfloor \gamma rank(W_0) \rfloor & \text{if } r > \gamma rank(W_0) \end{cases} \tag{8}$$

**Comparison and combination with LoRA.** PaRa has a corresponding relationship with the formulation of LoRA (cloneofsimo, 2022), $W_0 + \alpha BA$, i.e., we can consider $-Q$ as $B$ and $Q^T W_0$ as $A$. This means that we can combine individually trained LoRA and PaRa as $\Delta W = (-\alpha_1 Q + \alpha_2 B)(\alpha_1 Q^T W_0 + \alpha_2 A)$, where $\alpha_1$ and $\alpha_2$ are parameters controlling the strength of PaRa and LoRA. However, this combination method treats PaRa entirely as LoRA, failing to preserve control over the diversity of the generated images.

According to the rank-reduction property of PaRa, it does not require the scaling parameter. In other words, PaRa effectively determines the optimal scale $\alpha$, under which it reduces the rank. Therefore, we propose to combine the models as $W_{combine} = W_0 - QQ^T W_0 + \alpha_{LoRA} BA$. $\alpha_{LoRA}$ is the scale parameter of LoRA. In this way, we first reduce the rank and then add $BA$.

A question naturally arises that whether it is feasible to first apply LoRA and then reduce the rank of the new weights, *i.e.*, $W_{combine} = W_0 + \alpha_{LoRA}BA - QQ^T(W_0 + \alpha_{LoRA}BA)$. This is technically feasible but does not work well in practice since the variations introduced by LoRA extend the activation space of the initial layer, *e.g.*, learning a new concept on top of the existing data distribution. Placing PaRa afterwards would lead PaRa to attempt to eliminate these variations. The larger the intersection between the column vector spaces of PaRa's $Q$ and LoRA's $B$, the more the effect of LoRA will be diminished. An extreme example is that if $Q$ is the same as the orthogonal basis of $B$, then regardless of how $\alpha_{LoRA}$ is adjusted, the LoRA model will not have any effect. Comparative experiments can be found in Section 4.4.

### 3.3 Performing Single Image Editing Like Text-to-Image Generation

PaRa can perform single image editing directly through one-shot training and adjusting the text of the original training image. PaRa offers a solution by stabilizing the output, such that different Gaussian noises tend to yield the same result. This stability means that the model can generate images that closely resemble the training image, even when using various text prompts. This enables direct modification of the text prompt to facilitate image editing on the single training image.

By controlling the rank in PaRa, we can balance between faithful reconstruction and editability. When a large rank is selected, the model produces images that are very similar to the training image, enhancing reconstruction fidelity. Conversely, selecting a smaller rank increases the diversity of the generated images, improving editability. We provide a detailed mathematical discussion of how PaRa achieves this balance and the role of linear algebra concepts in Appendix F.

## 4 Experiments

The experiments evaluate PaRa on various tasks including single/multi-subject generation and single image editing, together with ablation studies. The SDXL1.0 (Podell et al., 2023) and the DDIM sampler with $\eta = 0$ are used for image generation. All experiments are conducted on a single A100 GPU with 40GB of VRAM.

### 4.1 Single-Subject Generation

**Implementation details.** We evaluated the effects of PaRa on customized single-subject generation, based on the Dreambooth dataset, where each label consists of five to six images. First, we verified that PaRa can indeed reduce the output space by confirming that the output diversity of PaRa has indeed decreased, compared to Vanilla SDXL1.0. Then we compared the effects of PaRa at different ranks with baseline methods on customized single-subject generation, including Dreambooth (Ruiz et al., 2023), Textual Inversion (Gal et al., 2022), OFT (Qiu et al., 2023), SVDiff (Han et al., 2023), LyCoris (Yeh et al., 2023) and LoRA (Hu et al., 2021; cloneofsimo, 2022). All baselines were trained for 1000 steps with a batch size of 1, and LoRA chose a rank of $r = 16$ and a scale of $\alpha = 2.2$ as the best model for a fair comparison. In our experiments, we found that PaRa already achieved ideal results at around 200 steps. Therefore, we compare the results of PaRa at 200 steps with the baselines at 1000 steps. Also, we employ a rank boundary $\gamma = 1/40$ for PaRa.

**Evaluation metrics.** We quantify the generation diversity using the metric of average SSIM. SSIM (Wang et al., 2004) is commonly used to measure image similarity. For a generated image set, we calculate the SSIM for each pair of images and then compute the average value. We use the CLIP score $cos(\tilde{\mathbf{x}}, c)$ to measure the text alignment between the generated image and the text (Radford et al., 2021), and we compute $1 - \mathcal{L}_{\text{LPIPS}}(\tilde{\mathbf{x}})$ to measure the image alignment between the generated image and the training image (Zhang et al., 2018), following the experimental design of SVDiff (Han et al., 2023) to create a comparison plot. Another benchmark we calculate for image alignment is the average CLIP image similarity scores $cos(\tilde{\mathbf{x}}, \mathbf{x})$ (Radford et al., 2021) between the training data and the generated image set (500 images). Here, the generated and training images are denoted as $\tilde{\mathbf{x}}$ and $\mathbf{x}$, respectively, and the prompt is denoted as $c$.

**Comparison of output diversity.** To verify that the diversity of the output has indeed decreased in PaRa, we compared the average SSIM of SDXL 1.0 (Podell et al., 2023) and its finetuned version

Table 1: Average SSIM results of two different prompts (top and bottom) with different subjects (each column) to indicate the diversity of the generated images. We compared Vanilla SDXL (Podell et al., 2023) with our PaRa using ranks 4 and 8. Higher average SSIM values suggest lower diversity and better alignment with the train images. We **bold** the results of the highest average SSIM values.

| "A [V] SCULPTURE MADE OF GOLD" | BEAR_PLUSHIE | CAT | DOG8 | DUCKTOY | GREY_SLOTH_PLUSHIE | MONSTER_TOY | RED_CARTOON | WOLF_PLUSHIE |
|---|---|---|---|---|---|---|---|---|
| VANILLA SDXL | 0.185±0.015 | 0.372±0.018 | 0.369±0.021 | 0.391±0.021 | 0.375±0.042 | 0.341±0.017 | 0.258±0.029 | 0.162±0.013 |
| PARA $r = 4$ | 0.203±0.008 | 0.390±0.033 | 0.463±0.011 | 0.516±0.010 | **0.382**±0.013 | 0.366±0.013 | 0.327±0.014 | **0.170**±0.012 |
| PARA $r = 8$ | **0.237**±0.010 | **0.401**±0.028 | **0.465**±0.020 | **0.519**±0.015 | 0.379±0.011 | **0.381**±0.015 | **0.386**±0.015 | 0.160±0.017 |

| "A [V] ON A SKATEBOARD IN TIMES SQUARE" | BEAR_PLUSHIE | CAT | DOG8 | DUCKTOY | GREY_SLOTH_PLUSHIE | MONSTER_TOY | RED_CARTOON | WOLF_PLUSHIE |
|---|---|---|---|---|---|---|---|---|
| VANILLA SDXL | 0.294±0.029 | 0.354±0.031 | 0.305±0.038 | 0.326±0.078 | 0.287±0.013 | 0.262±0.010 | 0.233±0.066 | 0.201±0.014 |
| PARA $r = 4$ | 0.327±0.025 | 0.403±0.082 | 0.307±0.025 | 0.351±0.025 | 0.374±0.023 | **0.276**±0.017 | 0.250±0.031 | 0.205±0.014 |
| PARA $r = 8$ | **0.339**±0.036 | **0.405**±0.110 | **0.323**±0.016 | **0.359**±0.031 | **0.383**±0.012 | 0.272±0.014 | **0.251**±0.044 | **0.232**±0.013 |

using PaRa. Based on 8 categories in Dreambooth, we generated 500 images for various prompts under different random seeds. Error bar is calculated by 5 runs. As shown in Table 1, on single-subject generation, PaRa demonstrates higher SSIM scores, indicating less diverse generations, compared to SDXL. Furthermore, as shown in Fig. 5, on single-image editing, this advantage becomes more apparent since it achieves even better average SSIM scores. Overall, it supports our assumption that the image space of the linear transformation $W_0 + \Delta W$ becomes smaller than that of $W_0$.

**Comparison of generation quality.** In Fig. 2, we compare PaRa with representative personalization techniques for T2I models, *i.e.*, LoRA (Hu et al., 2021) and SVDiff (Han et al., 2023). Overall, PaRa consistently achieves better image alignment performance across different prompts under different ranks. Moreover, in Fig. 3, we benchmark PaRa across all subjects and report the text alignment and image alignment scores under different ranks. It can be seen that PaRa results are positioned in the lower right part, indicating that PaRa achieves much better image alignment. Note that the relatively lower text-alignment ability of PaRa can be improved by reducing the rank $r$. In general, a larger value of $r$ usually helps to align the generated image with the learned concept, while a smaller value of $r$ allows the text prompt to more effectively control the generated image.

To quantitatively compare the fidelity of the generated images to the training images, we compared the average CLIP image similarity scores between PaRa and other personalization models, as shown in Table 2. Based on 8 categories in Dreambooth, we generated 500 images for the most basic prompt under different random seeds. The error bar is calculated from 5 runs. We compared LoRA with ranks 4 and 8 against PaRa with ranks 4 and 8. As shown in these 8 cases, PaRa achieved the highest scores in most instances, specifically in 7 out of 8 cases, mostly occurring at rank 8.

Table 2: CLIP image similarity scores between the training data and the generated image set (500 images). **Bold** values indicate the best scores.

| "A PHOTO OF [V]" | BEAR_PLUSHIE | CAT | DOG8 | DUCKTOY | GREY_SLOTH_PLUSHIE | MONSTER_TOY | RED_CARTOON | WOLF_PLUSHIE |
|---|---|---|---|---|---|---|---|---|
| PARA $r = 4$ | **0.8271** ± 0.0156 | 0.9315 ± 0.0101 | 0.8780 ± 0.0273 | 0.8913 ± 0.0277 | 0.8102 ± 0.0205 | 0.7627 ± 0.0282 | 0.7772 ± 0.0286 | 0.8516 ± 0.0261 |
| PARA $r = 8$ | 0.8051 ± 0.0209 | **0.9467** ± 0.0263 | **0.8955** ± 0.0225 | **0.8971** ± 0.0194 | **0.8384** ± 0.0186 | 0.7819 ± 0.0150 | 0.7955 ± 0.0262 | **0.8916** ± 0.0279 |
| LORA $r = 4$ | 0.7741 ± 0.0128 | 0.8057 ± 0.0241 | 0.7773 ± 0.0166 | 0.7935 ± 0.0124 | 0.7432 ± 0.0105 | 0.6930 ± 0.0182 | 0.6954 ± 0.0227 | 0.7359 ± 0.0164 |
| LORA $r = 8$ | 0.7943 ± 0.0260 | 0.8583 ± 0.0246 | 0.8295 ± 0.0113 | 0.8361 ± 0.0243 | 0.7672 ± 0.0122 | 0.7488 ± 0.0251 | 0.7462 ± 0.0274 | 0.7873 ± 0.0122 |
| SVDIFF | 0.7818 ± 0.0115 | 0.8854 ± 0.0254 | 0.8363 ± 0.0162 | 0.8439 ± 0.0252 | 0.7539 ± 0.0106 | 0.7558 ± 0.0146 | 0.7547 ± 0.0261 | 0.7035 ± 0.0146 |
| DREAMBOOTH | 0.7921 ± 0.0126 | 0.8893 ± 0.0148 | 0.8392 ± 0.0249 | 0.8520 ± 0.0123 | 0.7736 ± 0.0225 | 0.7813 ± 0.0281 | 0.7818 ± 0.0263 | 0.8067 ± 0.0169 |
| TEXTURE INVERSION | 0.7421 ± 0.0186 | 0.8048 ± 0.0227 | 0.7432 ± 0.0232 | 0.7798 ± 0.0166 | 0.7528 ± 0.0243 | 0.6225 ± 0.0274 | 0.7036 ± 0.0273 | 0.6836 ± 0.0193 |
| OFT | 0.8041 ± 0.0183 | 0.9024 ± 0.0282 | 0.8721 ± 0.0171 | 0.8865 ± 0.0111 | 0.7913 ± 0.0145 | 0.7808 ± 0.0211 | **0.7972** ± 0.0178 | 0.8346 ± 0.0136 |
| LYCORIS | 0.8092 ± 0.0214 | 0.9252 ± 0.0225 | 0.8637 ± 0.0136 | 0.8452 ± 0.0284 | 0.7874 ± 0.0154 | **0.7862** ± 0.0248 | 0.7589 ± 0.0217 | 0.8475 ± 0.0141 |

## 4.2 MULTI-SUBJECT GENERATION

We demonstrate the effectiveness of combining PaRas which are individually trained on different subjects. Mixing two T2I model weights in previous works (Wu et al., 2023; cloneofsimo, 2022) generally result in problematic generation results, where they tend to over-emphasize one subject or mix two subjects to a 'hybrid' entity that does not exist in reality. Moreover, these methods require auxiliary techniques such as Cut-Mix-Unmix data augmentation (Han et al., 2023), spatial condition (Gu et al., 2024), where the additional data annotation and training bring non-negligible financial or computational costs. In contrast, under our proposed PaRa, combining differently trained weights effectively eliminates the need for any auxiliary techniques while it can generate multiple subjects in a single image with little unrealistic blending.

The results shown in Fig. 4 were obtained using two challenging prompts. It can be observed that LoRA (Hu et al., 2021; cloneofsimo, 2022) only manages to generate the primary concept,

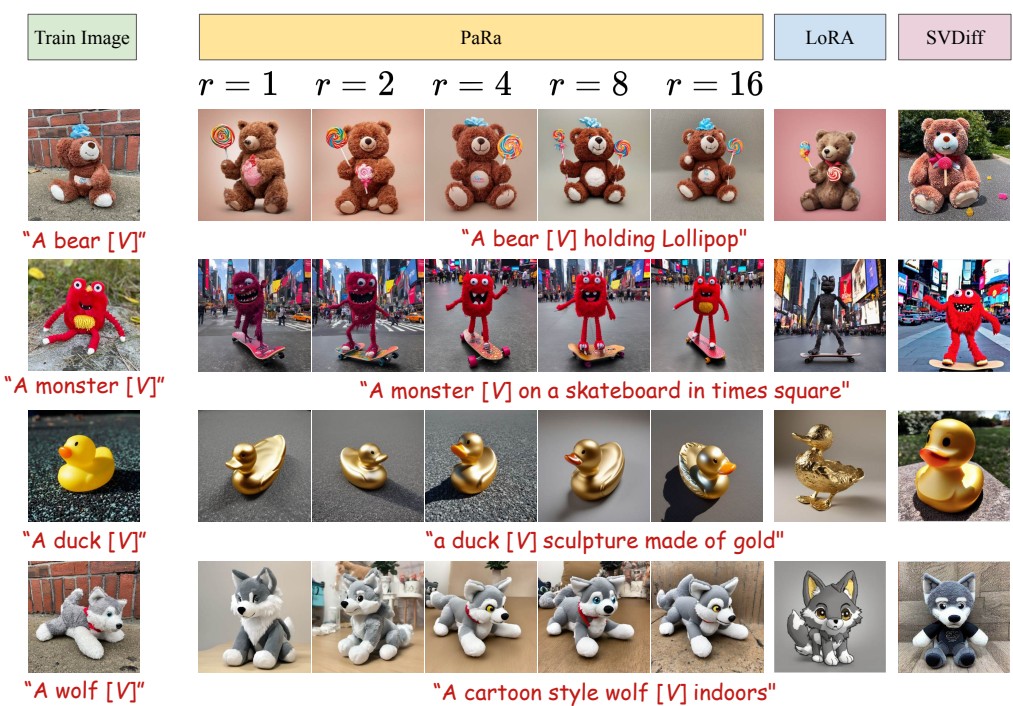

Figure 2: Comparing the proposed PaRa with LoRA (Hu et al., 2021) and SVDiff (Han et al., 2023) on single-subject generation. Each subject has 5 training images. PaRa includes results with ranks $r$ ranging from 1 to 16. For LoRA, we adopt a rank of 8. We provide more generation results on Dreambooth, Textual Inversion, OFT and LyCoris in Appendix E.

Concept1 [V1], into the generated images, while the second concept, Concept2 [V2], significantly deviates from the original image. In contrast, PaRa successfully captures both concepts well. As Cut-Mix-Unmix introduces unfair advantages due to the extra training data, we did not include it in our comparisons. More results can be found in Fig. 10 in Appendix D.

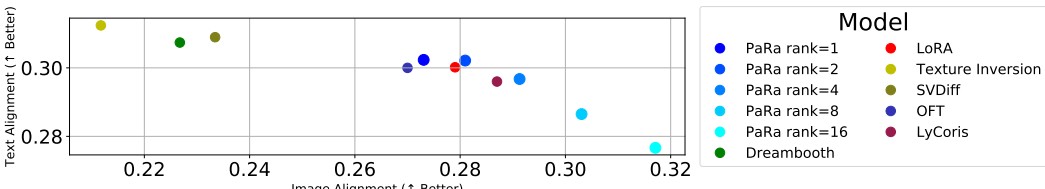

Figure 3: Text and image alignments for single subject generations. Text Alignment is measured by the CLIP score $cos(\tilde{\mathbf{x}}, c)$, and Image Alignmnet is measured by $1 - \mathcal{L}_{\text{LPIPS}}(\tilde{\mathbf{x}})$. The further to the right indicates higher fidelity to the personalized target, while higher along the vertical axis indicates improved text editability.

## 4.3 SINGLE IMAGE EDITING

In this section, we present the results of direct editing of a single image based on PaRa with $r = 8$. The experiment in Fig. 5 aims to demonstrate that with a properly chosen $r$, PaRa does not generate overly creative results and avoids the language drift issue (Ruiz et al., 2023). As a comparison, we also present the results of SVDiff (Han et al., 2023) on the same task without DDIM inversion. SVDiff can also partially achieve image editing effects without noise tracking (e.g. DDIM inversion). Here, we can see that PaRa has significantly less deviation from the original image compared to SVDiff. Numerically, for each target image and prompt pair, we generated 100 images and calculated their average SSIM, where the result of PaRa was significantly higher, reflecting the stability of the generated images.

## 4.4 COMBINATION WITH LoRA

In Fig. 6, we show the effects of combining PaRa and LoRA (Hu et al., 2021; cloneofsimo, 2022). We adopt public pre-trained LoRA models (mec, 2024; fig, 2024; dia, 2024) as examples to demonstrate the compatibility with PaRa weights. In general, PaRa complements existing LoRA weights as it naturally combines the strength of both models, *i.e*., a learned novel concept and image generation style. Here, we demonstrate our discussion in Section 3.2: during this combination, it is important to first apply PaRa and then LoRA. Conversely, applying LoRA first and then PaRa will, as shown in the figure, severely affect the performance of LoRA.

## 4.5 ABLATION AND OTHER ANALYSIS

**Parameter subsets and model sizes.** We evaluated using different subset parameters of UNet to update weights in Appendix B, which includes the model sizes of PaRa within different parameter subsets for finetuning and the corresponding generation performance.

**Rank bound $\gamma$.** In Section 3.2, we discuss the need for the rank boundary hyperparameter $\gamma$ to prevent the reduced rank from becoming too large, which could cause the generative model to collapse. In Section 4.2, we mention that our chosen rank boundary of $\gamma = 1/40$ is based on empirical results. In Appendix C, we provide such empirical results by comparinging the effects of different $\gamma$.

**Comparing different rank reductions $r$.** The effect of using different rank reductions $r$ in PaRa is one of the central topics of this paper. As $r$ increases, the diversity of the generated images decreases, leading to more faithful reconstructions in image editing; as $r$ decreases, the diversity of the generated images approaches that of the original models, improving text editability in image editing; if $r$ is too large, the generative model collapses. In Appendix D, we compare the performance of different $r$ values in image editing, multi-subject generation, and LoRA combination.

## 5 CONCLUSION

In conclusion, we have proposed PaRa, an effective and efficient framework for personalizing T2I diffusion models via parameter rank reduction. Our PaRa that finetunes with fewer parameters based on rank reduction can achieve better results than the prevailing LoRA fine-tuning. PaRa can well balance the diversity of generated images and the faithfulness to the customization objectives by selecting different ranks. PaRa is a flexible framework and performs well in different applications including single-subject generation, multi-subject generation, and single-image editing.

**Limitations.** PaRa focuses only on reducing the original output space. However, we acknowledge that because current diffusion models are not yet universal models, sometimes customization might still need to expand the original output space. Although we assume that as the generative model grows larger, the need for expansion will decrease, a model that can simultaneously address both space extension and space reduction may be stronger. There are algebraic problems concerning space extension hidden here that might be worth exploring.

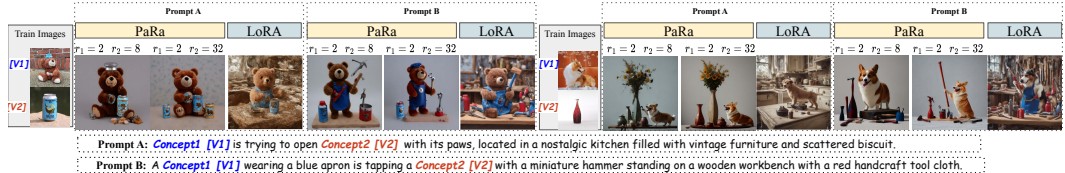

Figure 4: Examples of multi-subject generation results. The PaRa results are generated with the reduced rank $r_1 = 2$ for Concept 1 and $r_2$ being 8 and 32 for Concept 2. LoRA results are generated with the best rank of 16 for both concepts and the scale value of 1.

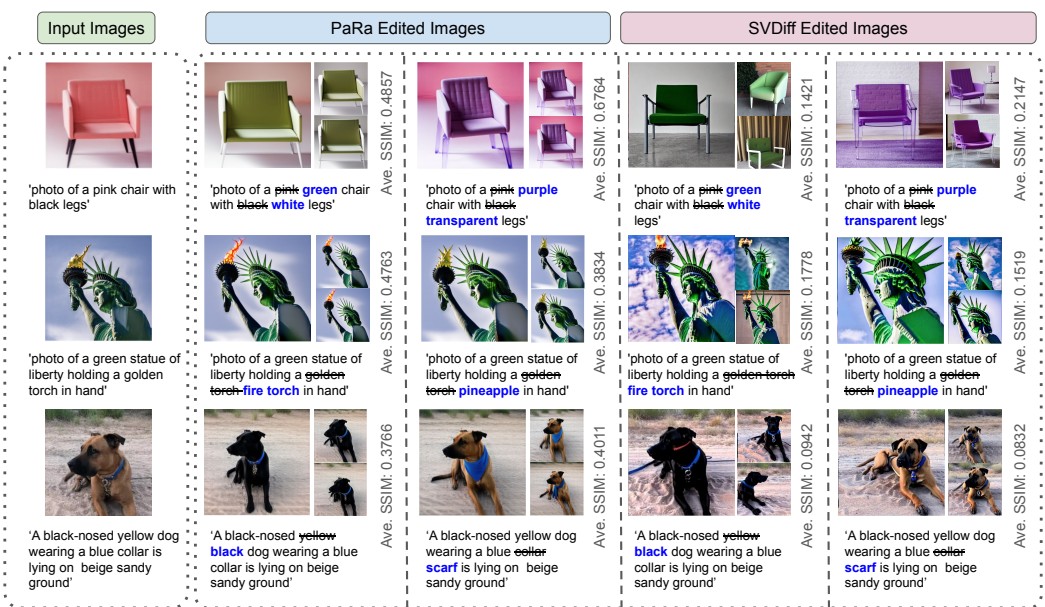

Figure 5: Single-image editing results. PaRa allows for image editing through one-shot learning of the original image and performs generation by directly modifying the prompt. We can see that PaRa achieves the expected modifications and preserves the personalized target subject well. In addition, PaRa achieves high consistency with untargeted elements of the initial image under different Gaussian noises.

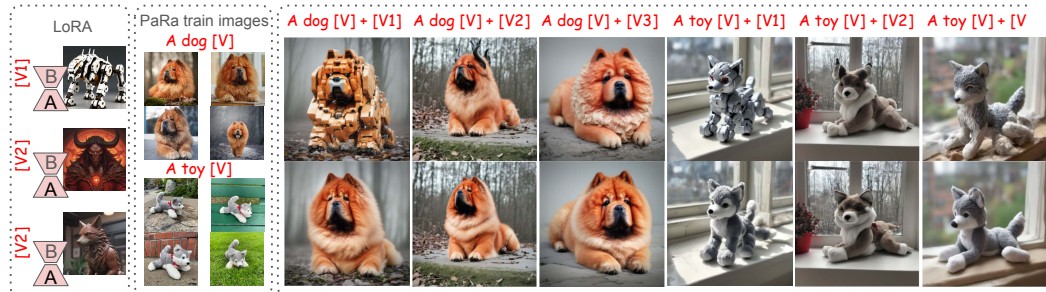

Figure 6: Generation results of combining PaRa with LoRA. Top: the results of first applying PaRa and then LoRA; Bottom: the results of first applying LoRA and then PaRa.

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

# Appendix

We organize our supplementary material as follows.

- In Section .1, we provide the theoretical basis for PaRa, including the proofs mentioned in Section 3.1.

- In Section A, we provide the proof mentioned in Section 3.2, explaining why PaRa Combination can be achieved through PaRa Sequential Addition.

- In Section B, we provide a comparison of the model sizes and generated results when different subsets of SDXL parameters are updated using PaRa.

- In Section C, we compare the generating results of selecting different Rank Boundaries $\gamma$.

- In Section D, we compare the performance of different $r$ values on the tasks of image editing and PaRa model combination.

- In Section E, we present additional generating results of PaRa, including comparisons with DreamBooth (Ruiz et al., 2023), Textual Inversion (Gal et al., 2022) and OFT (Qiu et al., 2023) on single-subject generation. We also provide more examples of effective PaRa combinations for multi-subject generation.

- In Section F, we provide a mathematical argument demonstrating why directly generating with PaRa in one-shot learning is an effective method for image editing, as a supplement to Section 3.3.

- In Section G, we present the results of a survey conducted to evaluate the quality of the generated images, with 209 participants.

- In Section H, we provide sufficient comparisons to demonstrate the improvement in training time achieved by our model.

- In Section I, we provide an overview of PaRa to assist readers who may not be familiar with the structural details of Stable Diffusion and similar models like LoRA and DreamBooth in understanding our work.

- In Section J, we present additional results of Single-image editing.

## .1 THEOREM AND PROOF OF PARA

**Theorem 1.** *For matrix $W$, the image space of $W$ is a d-dimension vector space $S_d$. If we have matrix $Q = [\vec{q_1}\ \vec{q_2}\ ...\ \vec{q_r}]$ with $q_i \in S_d$ and vectors $q_i$ are mutually orthonormal, then $W - QQ^TW$ has a (d-r)-dimension image space.*

This is actually an intuitive result, and to minimize any potential misleading, we provide a proof here.

**Proof**

Assume $W$ is a $b \times k$ matrix. Denote $W - QQ^TW$ as $[\vec{u_1}\ \vec{u_2}...\ \vec{u_k}]$. Denote $W$ as $[\vec{w_1}\ \vec{w_2}...\ \vec{w_k}]$. Present $W - QQ^TW$ in a vector form is

$$[\vec{u_1}\ \vec{u_2}...\ \vec{u_k}] = [\vec{w_1}\ \vec{w_2}...\ \vec{w_k}] - [\vec{q_1}\ \vec{q_2}...\ \vec{q_r}] \begin{bmatrix} \vec{q_1}^T \\ \vec{q_2}^T \\ ... \\ \vec{q_r}^T \end{bmatrix} [\vec{w_1}\ \vec{w_2}...\ \vec{w_k}]$$

$$= [\vec{w_1}\ \vec{w_2}...\ \vec{w_k}] - [\vec{q_1}\ \vec{q_2}...\ \vec{q_r}] \begin{bmatrix} \vec{q_1}^T\vec{w_1} & \vec{q_1}^T\vec{w_2} & ... & \vec{q_1}^T\vec{w_k} \\ \vec{q_2}^T\vec{w_1} & \vec{q_2}^T\vec{w_2} & ... & \vec{q_2}^T\vec{w_k} \\ ... & ... & ... & ... \\ \vec{q_r}^T\vec{w_1} & \vec{q_r}^T\vec{w_2} & ... & \vec{q_r}^T\vec{w_k} \end{bmatrix}$$

$$= [\vec{w_1}\ \vec{w_2}...\ \vec{w_k}] - \left[ \sum_{i=1}^{r} \vec{q_i}\vec{q_i}^T\vec{w_1} \quad \sum_{i=1}^{r} \vec{q_i}\vec{q_i}^T\vec{w_2} \quad ... \quad \sum_{i=1}^{r} \vec{q_i}\vec{q_i}^T\vec{w_k} \right]$$

For each column, it is $\vec{u}_j = \vec{w}_j - \sum_{i=1}^r \vec{q}_i \vec{q}_i^T \vec{w}_j$

The image space of $W$ is a d-dimension vector space $S_d$, then we have $d < b$ and $d < k$.

$(W - QQ^TW)$ has a rank $(d-r)$ means we need to show $\exists$ independent $\{\vec{g}_1, \vec{g}_2, ... \vec{g}_{d-r}\}, \vec{g}_i \in S_d$, such that $\forall \vec{u}_i \in \{\vec{u}_1, \vec{u}_2... \vec{u}_k\}, \exists$ scalars $a_1, ... a_{d-r} \in \mathbb{R}$, s.t. $\vec{u}_i = \sum_{i=1}^{d-r} a_i \vec{g}_i$.

Construct a new set $P$ containing all column vectors of the matrices $Q$ and $W$, $P = \{\vec{q}_1, \vec{q}_2, ..., \vec{q}_r, \vec{w}_1, \vec{w}_2, ..., \vec{w}_k\}$. Apply the Gram-Schmidt process to $P$, we will get $d$ orthogonal basis vectors $\{\vec{v}_1, \vec{v}_2, ..., \vec{v}_d\}$

Apply the Gram-Schmidt process to $P$:

Step1: $\vec{v}_1 = \vec{q}_1$

Step2: $\vec{v}_2 = \vec{q}_2 - \frac{\vec{q}_1 \cdot \vec{q}_2}{\vec{q}_1 \cdot \vec{q}_1} \vec{q}_1 = \vec{q}_2$

Step3: $\vec{v}_3 = \vec{q}_3 - \frac{\vec{q}_1 \cdot \vec{q}_3}{\vec{q}_1 \cdot \vec{q}_1} \vec{q}_1 - \frac{\vec{q}_2 \cdot \vec{q}_3}{\vec{q}_2 \cdot \vec{q}_2} = \vec{q}_3$

...

Step(r): $\vec{v}_r = \vec{q}_r$

Step(r+1): $\vec{v}_{r+1} = \vec{w}_1 - \sum_{i=1}^r \frac{\vec{q}_i \cdot \vec{w}_1}{\vec{q}_i \cdot \vec{q}_i} \vec{q}_i$

Step(r+2): $\vec{v}_{r+2} = \vec{w}_2 - \sum_{i=1}^r \frac{\vec{q}_i \cdot \vec{w}_2}{\vec{q}_i \cdot \vec{q}_i} \vec{q}_i - \frac{\vec{v}_1 \cdot \vec{w}_2}{\vec{v}_1 \cdot \vec{v}_1} \vec{v}_1$

...

As we only have $d$ basis vectors, we will have the orthogonal basis $\{\vec{q}_1, \vec{q}_2, ..., \vec{q}_r, \vec{v}_{r+1}, ..., \vec{v}_d\}$ finally.

Assume $\vec{w}_j = \sum_{l=1}^r a_l \vec{q}_l + \sum_{l=r+1}^d b_l \vec{v}_l$

$\vec{u}_j = \vec{w}_j - \sum_{i=1}^r \vec{q}_i \vec{q}_i^T \vec{w}_j$
$= \sum_{l=1}^r a_l \vec{q}_l + \sum_{l=r+1}^d b_l \vec{v}_l - \sum_{i=1}^r \vec{q}_i \vec{q}_i^T (\sum_{l=1}^r a_l \vec{q}_l + \sum_{l=r+1}^d b_l \vec{v}_l)$
$= \sum_{l=1}^r a_l \vec{q}_l + \sum_{l=r+1}^d b_l \vec{v}_l - \sum_{l=1}^r a_l \vec{q}_l$ (as $\vec{q}_i^T \vec{q}_l = 1$ if $i = l$, $\vec{q}_i^T \vec{q}_l = 0$ if $i \neq l$, $\vec{q}_i^T \vec{v}_l = 0$ if $l \geq r+1$)
$= \sum_{l=r+1}^d b_l \vec{v}_l$

As we want to proof $\exists \{\vec{g}_1, \vec{g}_2, ... \vec{g}_{d-r}\}$, s.t. $\vec{g}_i \in S_d, \forall \vec{u}_i \in \{\vec{u}_1, \vec{u}_2... \vec{u}_k\}, \exists \{a_1, ... a_{d-r}\}$, s.t. $\vec{u}_i = \sum_{i=1}^{d-r} a_i \vec{g}_i, a_i \in \mathbb{R}$,

so we have $\{\vec{g}_1, \vec{g}_2, ... \vec{g}_{d-r}\} = \{\vec{v}_{r+1}, ..., \vec{v}_d\}$, s.t. $\vec{g}_i \in S_d, \forall \vec{u}_i \in \{\vec{u}_1, \vec{u}_2... \vec{u}_k\}, \exists \{a_1, ... a_{d-r}\}$, s.t. $\vec{u}_i = \sum_{i=1}^{d-r} a_i \vec{g}_i, a_i \in \mathbb{R}$

Therefore, $W - QQ^TW$ has a (d-r)-dimension image space.

**Note:** The case not covered by this theorem is when there exists $Q$'s column vector $\vec{q}_l \notin S_d$. When $\vec{q}_l \notin S_d$, we have $\vec{q}_l \cdot \vec{w}_j = 0$ for any column vector $\vec{w}_j$ of $W$. That is, this component $\vec{q}_l$ will not update $W$ in $W - QQ^TW$, because $\vec{w}_j - \vec{q}_l \vec{q}_l^T \vec{w}_j = 0$. In PaRa, if the PaRa training has not yet converged, then this $\vec{q}_l$ will be updated. If it has already converged, it indicates that our choice of $r$ might be too large, leading to over-parameterization of PaRa, but the other $\vec{q}_i \in S_d$ can still achieve the objective of personalization.

## A    PROOF OF PARA SEQUENTIAL ADDITION IN PARA COMBINATION

Here we will prove that in PaRa combination, $W_0 - Q'_m Q'^T_m W_0 = W_1 - Q_2 Q_2^T W_1 = W_0 - Q_1 Q_1^T W_0 - Q_2 Q_2^T W_0 + Q_2 Q_2^T Q_1 Q_1^T W_0$.

For two trained PaRa models, we have parameters:

$Q1 = [\vec{q}_{11} \ \vec{q}_{12} \ ... \vec{q}_{1r_1}]_{d \times r_1}$

$Q2 = [\vec{q}_{21} \ \vec{q}_{22} \ ... \vec{q}_{2r_2}]_{d \times r_2}$.

For the standard combination of $Q_1$ and $Q_2$: $Qm = [\vec{q}_{11}\ \vec{q}_{12}\ ...\vec{q}_{1r_1}\vec{q}_{21}\ \vec{q}_{22}\ ...\vec{q}_{2r_2}]_{d\times(r_1+r_2)}$

With $Q'_m R'_m = Q_m$, same as what we dicussed in Appendix A, we can find out a orthonormal basis $\{\vec{q}_{11}, \vec{q}_{12}...\vec{q}_{1r_1}, \vec{p}_1, \vec{p}_2...\vec{p}_{(r_m-r_1)}\}$ of $r_m$ vectors from the column vectors of $Q_1$ and $Q_2$.

$W_m = W_0 - Q'_m Q'^T_m W_0$ means for each column vector $\vec{w}_i$ of $W$, it is transferred to

$$\vec{w}_i - \sum_j^{r_1}(\vec{q}_{1j}\cdot\vec{w}_i)\vec{q}_{1j} - \sum_l^{r_m-r_1}(\vec{p}_l\cdot\vec{w}_i)\vec{p}_l \tag{9}$$

For the practical combination $W_m = W_1 - Q_2 Q_2^T W_1 = W_0 - Q_1 Q_1^T W_0 - Q_2 Q_2^T W_0 + Q_2 Q_2^T Q_1 Q_1^T W_0$, $\vec{w}_i$ is transferred to

$$\vec{w}_i - \sum_j^{r_1}(\vec{q}_{1j}\cdot\vec{w}_i)\vec{q}_{1j} - \sum_s^{r_2}(\vec{q}_{2s}\cdot\vec{w}_i)\vec{q}_{2s} + \sum_s^{r_2}((\sum_j^{r_1}(\vec{q}_{1j}\cdot\vec{w}_i)\vec{q}_{1j})\cdot\vec{q}_{2s})\vec{q}_{2s} \tag{10}$$

We have $\vec{q}_{2s} = \sum_j^{r_1} a_{sj}\vec{q}_{1j} + \sum_l^{r_m-r_1} b_{sl}\vec{p}_l$, and $a_{sj} = \vec{q}_{1j}\cdot\vec{q}_{2s}$, $b_{sl} = \vec{p}_l\cdot\vec{q}_{2s}$ then expression 10 is

$$\vec{w}_i - \sum_j^{r_1}(\vec{q}_{1j}\cdot\vec{w}_i)\vec{q}_{1j} - \sum_s^{r_2}(\sum_j^{r_1} a_{sj}\vec{q}_{1j} + \sum_l^{r_m-r_1} b_{sl}\vec{p}_l\cdot\vec{w}_i)\vec{q}_{2s} + \sum_s^{r_2}((\sum_j^{r_1}(\vec{q}_{1j}\cdot\vec{w}_i)\vec{q}_{1j})\cdot\vec{q}_{2s})\vec{q}_{2s} \tag{11}$$

$$= \vec{w}_i - \sum_j^{r_1}(\vec{q}_{1j}\cdot\vec{w}_i)\vec{q}_{1j} - \sum_s^{r_2}((\sum_j^{r_1} a_{sj}\vec{q}_{1j} + \sum_l^{r_m-r_1} b_{sl}\vec{p}_l)\cdot\vec{w}_i)\vec{q}_{2s} + \sum_s^{r_2}((\sum_j^{r_1}(\vec{q}_{1j}\cdot\vec{w}_i)\vec{q}_{1j})\cdot\vec{q}_{2s})\vec{q}_{2s} \tag{12}$$

$$= ... - \sum_s^{r_2}((\sum_j^{r_1} a_{sj}\vec{q}_{1j}\cdot\vec{w}_i) - ((\sum_j^{r_1}(\vec{q}_{1j}\cdot\vec{w}_i)\vec{q}_{1j})\cdot\vec{q}_{2s}))\vec{q}_{2s} + \sum_s^{r_2}\sum_l^{r_m-r_1}(b_{sl}\vec{p}_l\cdot\vec{w}_i)\vec{q}_{2s} \tag{13}$$

When substituting $a_{sj}$ and $b_{sl}$, the intermediate terms are eliminated, which is

$$\vec{w}_i - \sum_j^{r_1}(\vec{q}_{1j}\cdot\vec{w}_i)\vec{q}_{1j} + \sum_s^{r_2}\sum_l^{r_m-r_1}(\vec{p}_l\cdot\vec{q}_{2s}\vec{p}_l\cdot\vec{w}_i)\vec{q}_{2s} \tag{14}$$

$$= \vec{w}_i - \sum_j^{r_1}(\vec{q}_{1j}\cdot\vec{w}_i)\vec{q}_{1j} + \sum_l^{r_m-r_1}\sum_s^{r_2}(\vec{p}_l\cdot\vec{q}_{2s}\vec{q}_{2s})\vec{p}_l\cdot\vec{w}_i \tag{15}$$

The term $\sum_s^{r_2}(\vec{p}_l\cdot\vec{q}_{2s}\vec{q}_{2s})$ represents the total component of $\vec{p}_l$ in the column space of $Q_2$. As $\vec{p}_l$ is orthogonal with $\{\vec{q}_{11}, \vec{q}_{12}...\vec{q}_{1r_1}\}$, we have $\sum_s^{r_2}(\vec{p}_l\cdot\vec{q}_{2s}\vec{q}_{2s}) = \vec{p}_l$. Hence, it is proven that Expression 9 and Expression 15 are equal.

## B    COMPARISONS OF THE GENERATION PERFORMANCE FOR DIFFERENT SUBSET PARAMETERS

ExCA, as a parameter subset, is chosen in our experiments. Here, we provide additional examples generated using other subsets of parameters. We experimented with eight different parameter subsets.

- **Exclude Cross-Attention (ExCA)**
  - This method trains all layers except cross-attention and time embedding layers.
- **Exclude Self-Attention (ExSA)**
  - This method trains all layers except self-attention layers.
- **Self-Attention Only (SAO)**

- This method trains only the self-attention layers.
- **Cross-Attention Only (CAO)**
    - This method trains only the cross-attention layers.
- **Full Model Training (FMT)**
    - This method trains all layers of the model.
- **Strict Cross-Attention (SCA)**
    - This method trains only the queries and keys within the cross-attention mechanisms.
- **Exclude Cross-Attention High-Level (ExCA-HL)**
    - This method trains all layers except cross-attention layers, with an emphasis on high-level feature representations.
- **Exclude Cross-Attention High-Level Last (ExCA-HL-Last)**
    - This method trains all layers except cross-attention layers, focusing on the final stages of the high-level feature space.

As shown in Table 3, regardless of the subset used, the parameter count of PaRa is significantly lower than that of LoRA. In Fig. 7, we use the example of a bear plushie, with $r = 16$, to illustrate the effects of different parameter subsets. It can be observed that models with larger parameter counts, such as FMT, CAO, and ExSA, tend to align better with the training images. Conversely, models with smaller parameter counts may not align well with the target subject but match the text more closely. The model with the largest parameter count, FMT, even produced a 'hybrid' result in the multi-subject example "A girl is holding a small bear [V]". Models like SAO and SCA strike a better balance. EXCA, being a well-performing subset, has numerous examples listed in other sections and will not be repeated here.

| | $r=2$ | | $r=16$ | | $r=128$ | |
| SUBSET | PARA | LORA | PARA | LORA | PARA | LORA |
|---|---|---|---|---|---|---|
| EXCLUDE CROSS-ATTENTION (EXCA) | 1.8 MB | 4.8 MB | 13 MB | 33 MB | 87 MB | 190 MB |
| EXCLUDE SELF-ATTENTION (EXSA) | 1.9 MB | 4.8 MB | 13 MB | 33 MB | 87 MB | 190 MB |
| SELF-ATTENTION ONLY (SAO) | 1.6M | 3.1M | 11M | 21M | 82M | 163M |
| CROSS-ATTENTION ONLY (CAO) | 1.6M | 3.5M | 11M | 25M | 82M | 193M |
| FULL MODEL TRAINING (FMT) | 3.4M | 8.2M | 23M | 58M | 169M | 382M |
| STRICT CROSS-ATTENTION (SCA) | 1.2M | 2.8M | 7.9M | 20M | 62M | 152M |
| EXCLUDE CROSS-ATTENTION HIGH-LEVEL (EXCA-HL) | 276K | 700K | 1.9M | 5.0M | 14M | 30M |
| EXCLUDE CROSS-ATTENTION HIGH-LEVEL LAST (EXCA-HL-LAST) | 6.9K | 53K | 42K | 403K | 82K | 803K |

Table 3: Fine-tuning subsets of parameters in UNet, comparing them with LoRA at various ranks, along with their corresponding model sizes.

## C   COMPARISON OF DIFFERENT RANK BOUNDARIES $\gamma$

In Section 3.2, we discussed that setting the $r$ too high can lead to model instability. To mitigate this issue, it is necessary to establish a rank boundary $\gamma$. We empirically chose $\gamma = \frac{1}{40}$ across various experimental scenarios. The Fig. 8 illustrates the results generated with different $\gamma$ values. Red markers indicate errors in the generated images. The figure demonstrates that when $r$ is small, the setting of $\gamma$ has minimal impact. However, as $r$ increases, it becomes crucial to select an appropriate $\gamma$, with values around $\frac{1}{20}$ to $\frac{1}{40}$ yielding the best performance.

## D   DIFFERENT $r$ VALUES IN IMAGE EDITING AND MODEL COMBINATION

In Figure 9, we compare the performance of PaRa in one-shot learning for image editing across different rank values. We also calculate the average SSIM for each prompt generated with different ranks. It can be observed that the results for $r = 2$ to $r = 8$ show minimal differences. In such cases, it is natural to prefer $r = 2$, as it requires fewer parameters and reduces computational complexity while maintaining comparable results. However, when $r$ is larger, such as $r = 16$ to $32$, the generated results for more challenging images tend to degrade significantly.

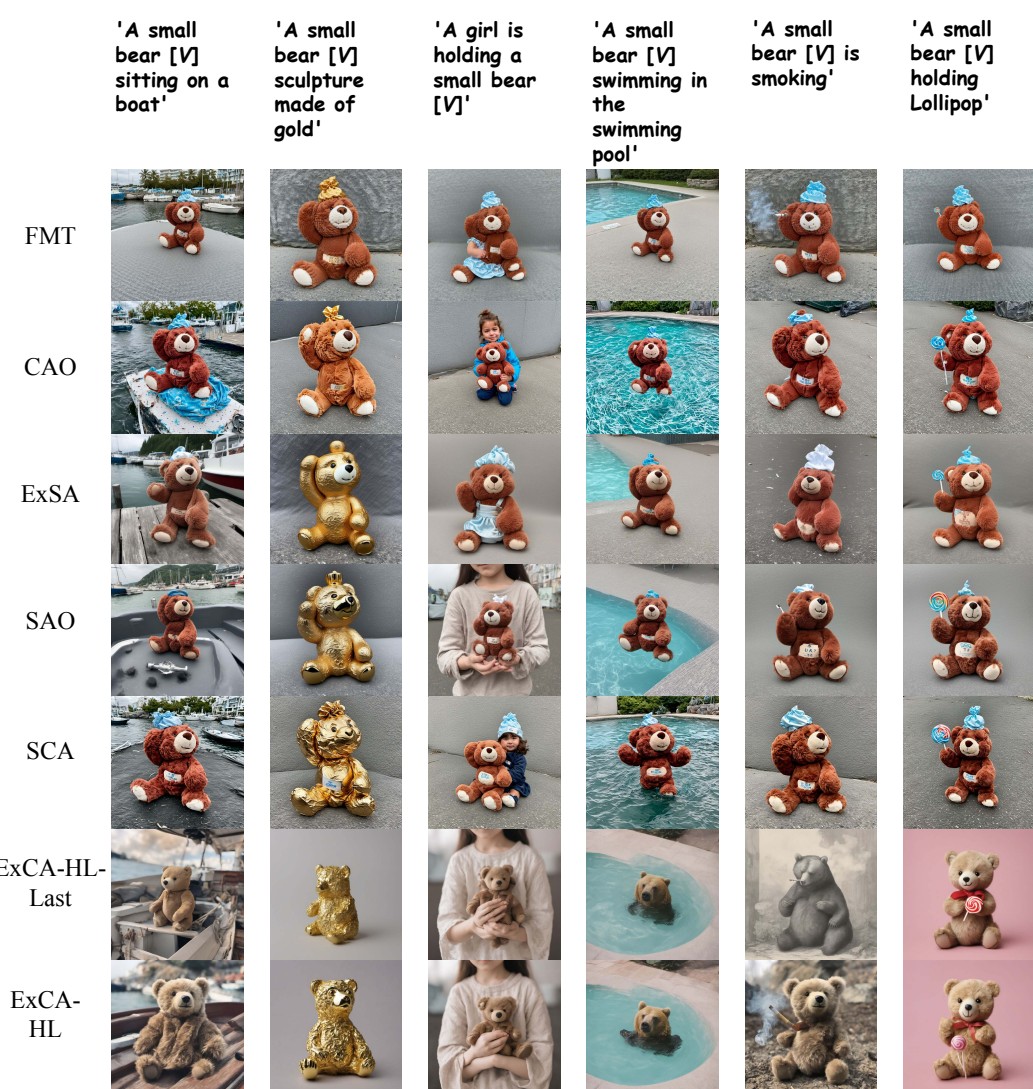

Figure 7: PaRa Single subject generation on different parameter subsets

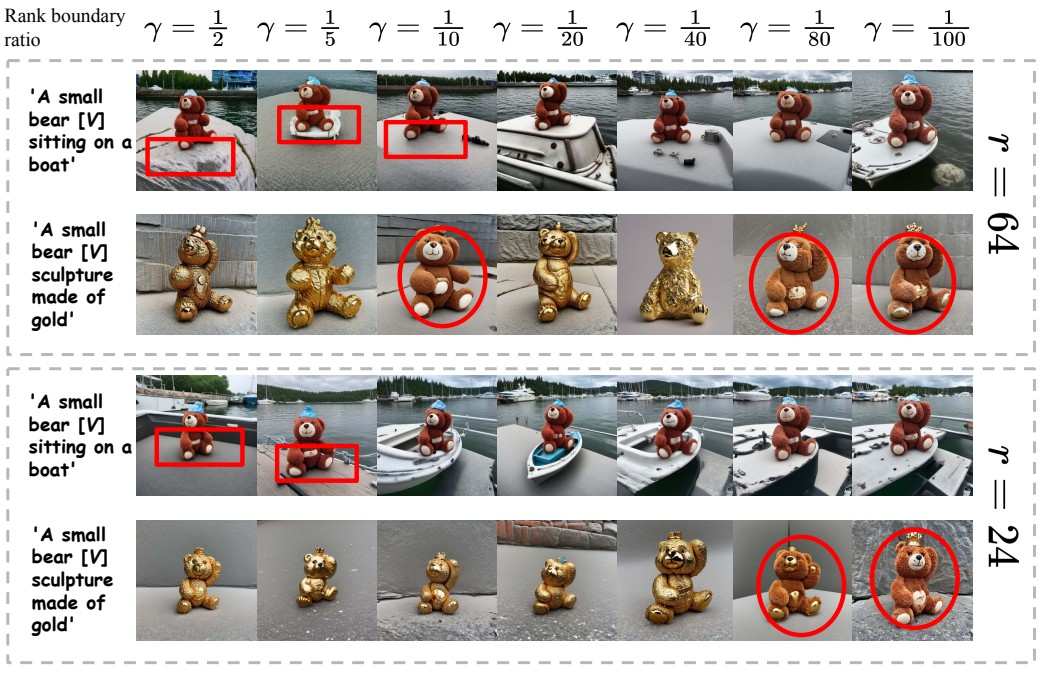

Figure 8: Comparison of Different Rank Boundaries $\gamma$ for single subject generation. Red boxes indicate errors in generating "boat", and red circles indicate errors in generating "gold".

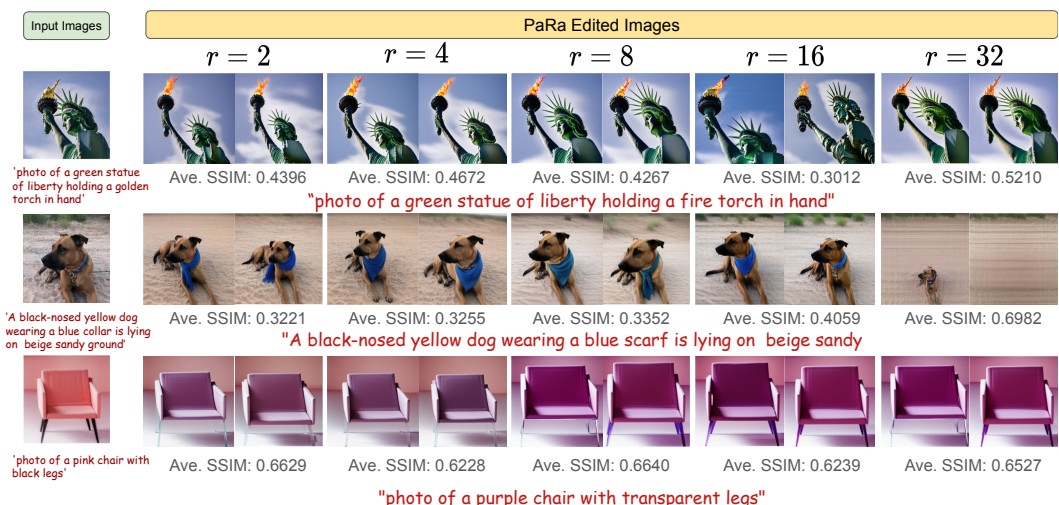

Figure 9: Comparison of Different Rank Image Editing. $r = 2$ to $r = 8$ show minimal differences. When $r$ is larger, such as $r = 16$ to $32$, the generated results for more challenging images tend to degrade significantly.

In Figure 10, we compare the performance of PaRa with different ranks in model combination. The results indicate that ranks between 2 and 8 are more suitable. When the rank $r$ is too large, the interaction between the two PaRa models becomes significant, potentially causing one subject to be ignored in the generated results. However, if there is a primary and secondary subject, the rank $r$ of the PaRa corresponding to the primary subject can be set higher. For example, in Figure 10, the dog and bear are considered primary subjects, so setting $r = 8$ is appropriate to capture their details. On the other hand, for secondary subjects such as the can or vase, it is preferable to keep $r$ as low as 2. This setting can be applied to the general combination of primary and secondary objects.

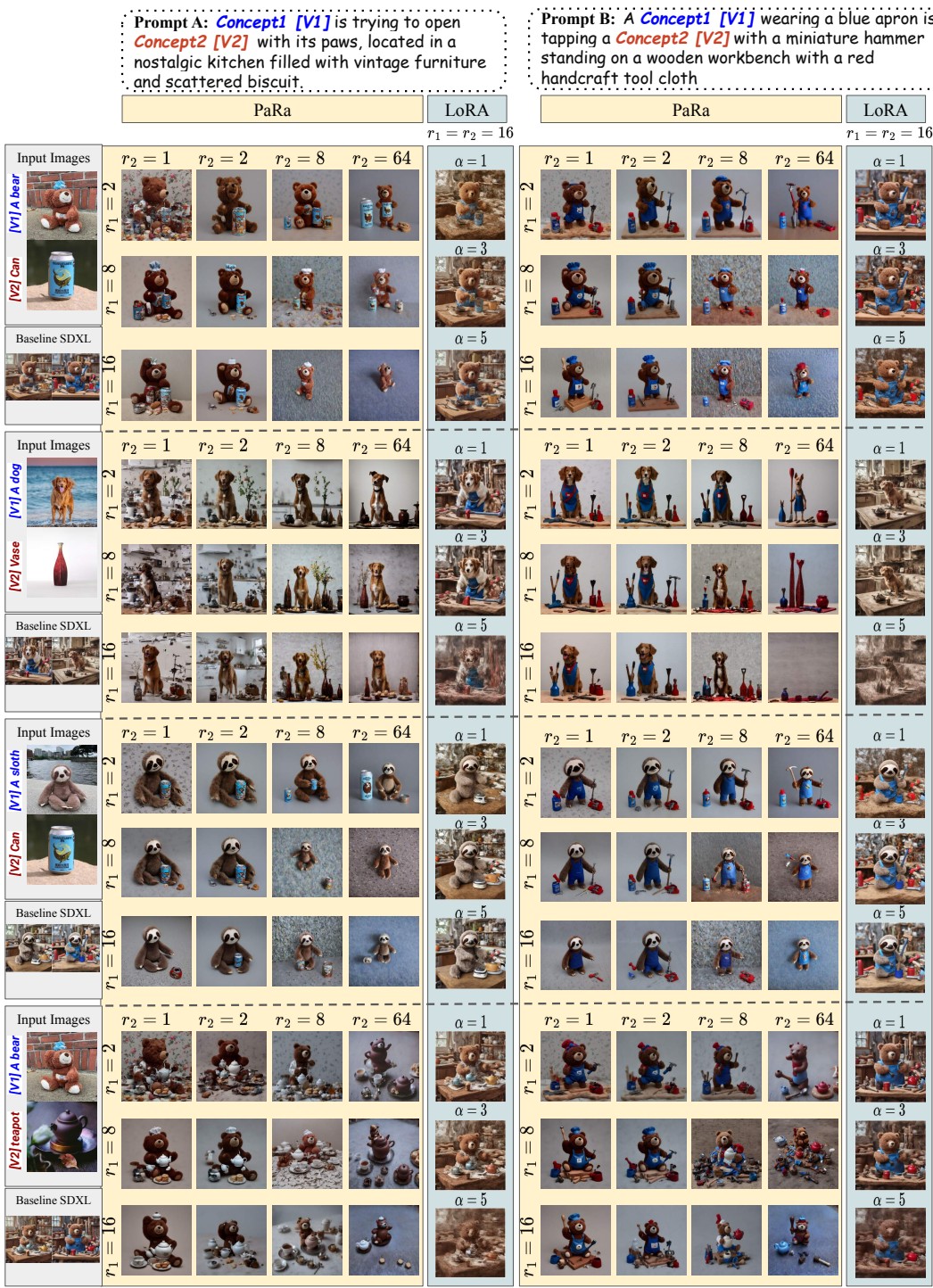

Figure 10: Multi-Subject Generation. In the PaRa comparison, we analyzed results for Concept 1 with reduced ranks $r_1$ set at 2, 8, 16; and for Concept 2, the reduced ranks $r_2$ were 1, 2, 8, 64. LoRA, based on experimental optimization, used the best rank of 16 for both concepts and compared different scales with alpha values of 1, 3, 5.

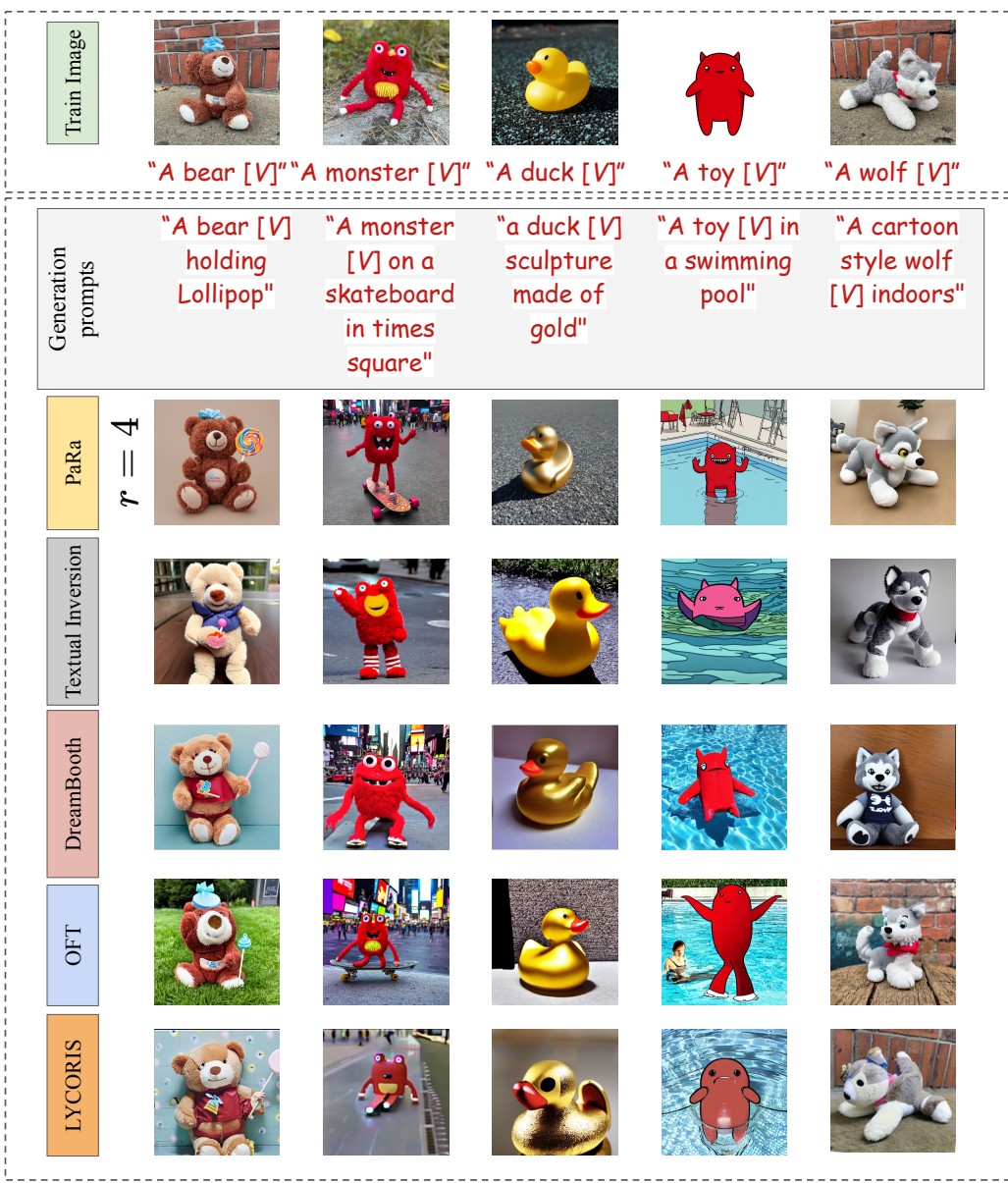

Figure 11: Comparison of Single-Subject Generation with DreamBooth, Textual Inversion, OFT and LyCoris

# E  MORE GENERATION RESULTS

In Figure 11, we compare the performance of PaRa in single-subject generation with DreamBooth, Textual Inversion and OFT. The results clearly demonstrate the significant advantages of PaRa.

In Figure 12, we present more examples of PaRa combinations. In these examples, the primary subject rank is 2, and the secondary subject rank is 32, which we have found through experience to be a relatively suitable pairing.

| | BEAR_PLUSHIE | RED_TOY | DUCKTOY | RED_CARTOON | WOLF_PLUSHIE | BEAR_PLUSHIE&TIN | DOG&VASE | CAT&TEAPOT | GREY_SLOTH_PLUSHIE&TIN |
|---|---|---|---|---|---|---|---|---|---|
| HUMAN PREFERENCE PARA | 96.17% | 97.13% | 91.18% | 91.90% | 96.62% | 82.08% | 89.94% | 78.45% | 88.76% |
| HUMAN PREFERENCE LORA | 3.83% | 2.87% | 8.82% | 8.09% | 3.38% | 7.54% | 3.91% | 15.47% | 7.30% |

Table 4: Human evaluation results comparing PaRa and LoRA. The first five columns compare single-subject generation, while the last four columns compare multi-subject combination. Each of the last four columns does not sum to $100\%$ because there is a "Neither" option in the survey indicating dissatisfaction with both results.

## F    IMAGE EDITING ALGEBRAIC DISCUSSION

The stability of PaRa outputs is reflected in that different Gaussian noises tend to yield the same result, represented as:

$$h = Wx = W(x + \Delta x) \tag{16}$$

the equation is true when $W\Delta x = \mathbf{0}$, which is denoted by $\Delta x \in kernel(W)$.

According to the rank-nullity theorem, for the linear transformation $W : X \rightarrow H$, $rank(W) + nullity(W) = \dim X$. (The nullity of $W$ is the dimension of $W$ kernel).) In PaRa, our reduced rank $r = rank(W_0) - rank(W_{reduce}) = nullity(W_{reduce}) - nullity(W_0)$, we have $rank(W_0) > rank(W_{reduce})$ which implies $nullity(W_{reduce}) > nullity(W_0)$, more of $\Delta x$ in PaRa will not produce different outputs. This reduced rank $r$ demonstrates the problem (Meng et al., 2021) of the trade-off between faithful reconstruction and editability in image editing, As $r$ increases, the modifiable features decrease, making the reconstruction more faithful. As $r$ decreases, the modifiable features increase, bringing the diversity of the generated images closer to that of the underlying pre-trained generation model, and improving the editability. When a large $r$ is selected for training the PaRa on a single image, the model generates images that closely resemble the training image, even when using various text prompts. This enables direct modification of the text prompt to facilitate image editing on the single train image.

## G    SURVEY: USER ASSESSMENT OF SINGLE-SUBJECT AND MULTI-SUBJECT IMAGE GENERATION

To evaluate the quality of the generated images, we conducted a survey where 209 participants provided their feedback. In order to streamline and ensure effectiveness, we designed a survey form consisting of nine questions. These questions allowed users to compare the image generation performance of PaRa and LoRA, covering both single-subject and multi-subject generation, as well as comparisons across different ranks.

The survey form is shown in this link:

```
https://docs.google.com/forms/d/e/1FAIpQLSc7mDMagFkotVwYiYjftV3FY_
WSoYDtcpX6_3VqVlp9SmREbA/viewform?usp=sf_link.
```

If you click on the survey link, you will find that participants are blind to which model corresponds to each image, which helps reduce subjective bias. The survey may have regional bias because 181 participants completed the Google Form, while 28 participants completed the Tencent Form. The survey consists of 9 questions. The first five questions are actually about PaRa and LoRA's Single-Subject generation. Options (a)-(e) correspond to PaRa ranks 1, 2, 4, 8, and 16, respectively. Options (f) and (g) correspond to LoRA rank 8 with scales 1.0 and 2.2, respectively. The last four questions are about multi-subject generation, where options (a) and (b) correspond to the results of PaRa and LoRA, respectively.

The overall results are summarized in Table 4.

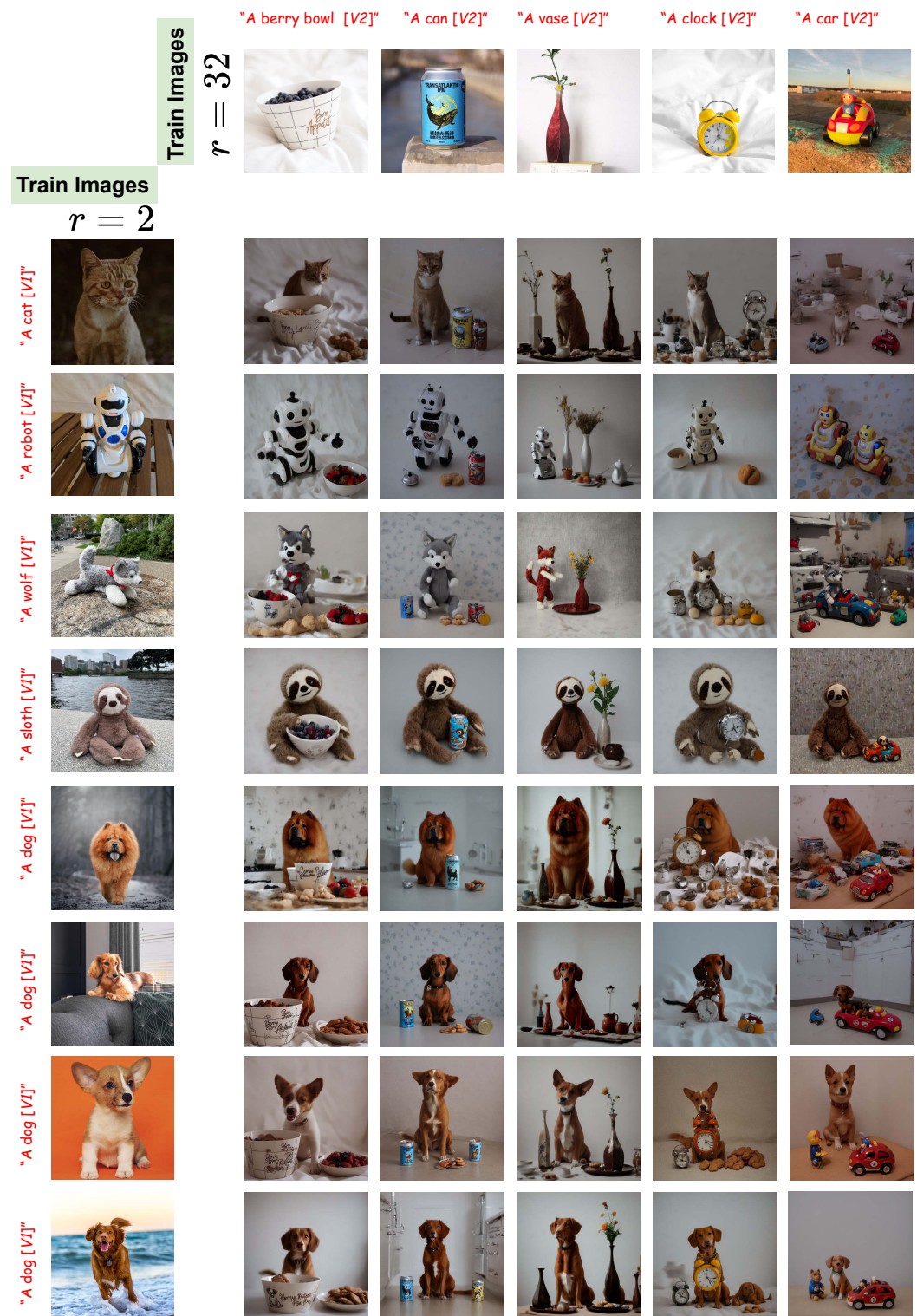

Figure 12: More Multi-Subject Generation Examples: The left column shows the training images of the primary subject, and the top row shows the training images of the secondary subject, each represented by a single image but actually based on five images for few-shot training. The prompts used here are "[V1] in the kitchen with a [V2] next to it."

|  | $r$=2 | | $r$=16 | | $r$=32 | | $r$=128 | |
|---|---|---|---|---|---|---|---|---|
| MODEL | PARA | LORA | PARA | LORA | PARA | LORA | PARA | LORA |
| MODEL SIZE | 1.8 MB | 4.8 MB | 13 MB | 33 MB | 22 MB | 56 MB | 87 MB | 190 MB |
| TRAIN. TIME | 5.8MIN | 6.2MIN | 6.9MIN | 10.4MIN | 9.4MIN | 15.1MIN | 11.2MIN | 20.1MIN |

Table 5: Training time and model size comparison between LoRA and PaRa.

## H    HOW EFFICIENT IS PARA

In the Table 5, we compare the model size and training time of PaRa and LoRA when training for 1000 steps using the subset parameters of ExCA, as described in Appendix Section B. It can be observed that as the selected rank increases, the training time advantage of PaRa becomes more pronounced.

## I    VISUAL OVERVIEW OF THE PARA FRAMEWORK

Figure 13 presents an overview of the PaRa framework. It follows a process similar to LoRA but focuses on the subspace of the entire model's output. For the linear weights $W$ in the U-net of the diffusion model, such as the $QKV$ units in the attention module of the Unet, originally $W$ serves as a linear transformation, and its image space is $S_h$. If each column vector of $W$ is subtracted by its component on a base vector, resulting in the reduction of $W$'s rank to $W_{reduce}$, the dimension of $S_h$ will decrease. This ultimately reduces the diversity of the final output of the generative model.

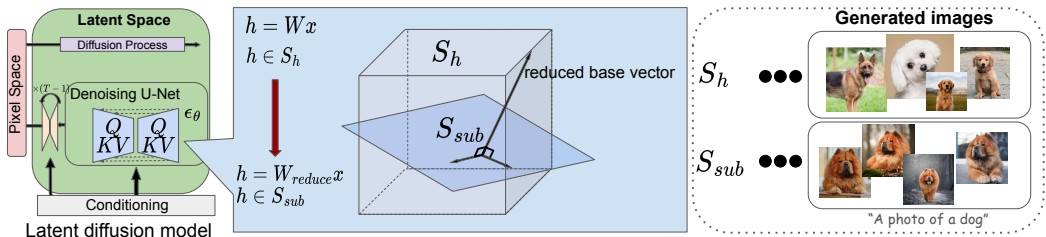

Figure 13: Overview of the PaRa framework

## J    ADDITIONAL EXAMPLES OF PARA IMAGE EDITING PERFORMANCE

Figure 14 provides additional examples showcasing PaRa's performance on different one-shot pairs derived from subjects in the DreamBooth dataset. These examples highlight the model's robustness across diverse subjects and varying input pairs.

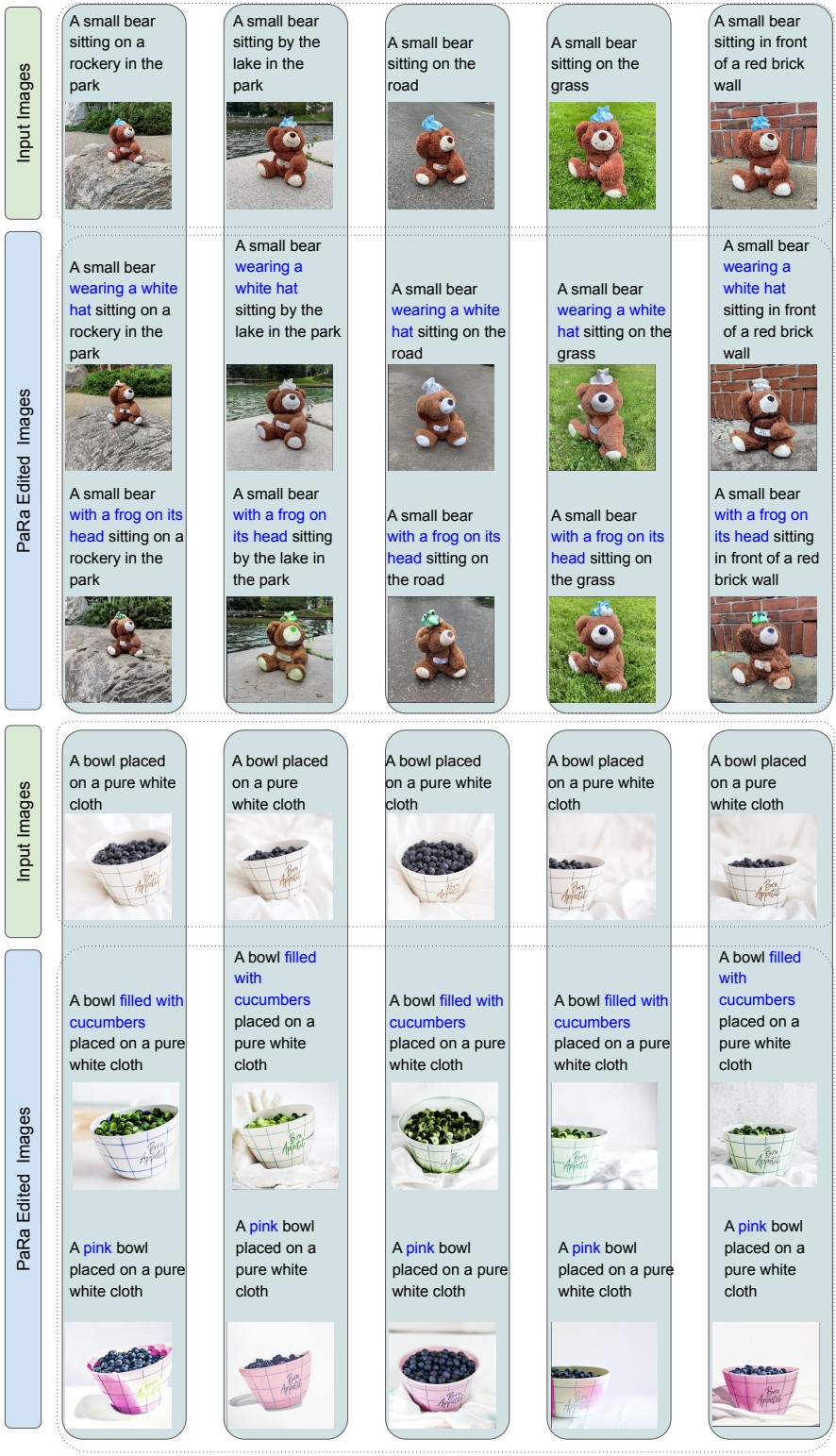

Figure 14: More single-image editing results: The top and bottom parts represent two different subjects, with input images sourced from the DreamBooth dataset. Each column shows the training input image and its corresponding edited result. The prompts used for PaRa-edited images are displayed above each edited image.

