# OpenReview forum: "PaRa: Personalizing Text-to-Image Diffusion via Parameter Rank Reduction"
_ICLR.cc/2025/Conference — ICLR 2025 Spotlight_

### Official Review · Reviewer_PSkP · 2024-11-01

**Soundness:** 3
**Presentation:** 3
**Contribution:** 3
**Rating:** 8
**Confidence:** 3

**Summary:**

This paper proposes a new method to personalizing diffusion model with few samples, which is to reduce the rank of the original weights in diffusion models. This method allows the finetuned model to have smaller diversity and better alignment with the training data. This paper conducts extensive experiments on various generation task including single subject, multi-subject generation and editing. The paper is well written and easy to understand.

**Strengths:**

1) This paper provides a very interesting ideas to personalize diffusion model by reducing the rank of the weights.
2) The proposed finetuning method is simple, effective and show good flexibility.

**Weaknesses:**

1) Different hyper-parameters may affect the alignment of the previous methods, such as the Rank of LoRA and fine-tuning steps. Authors may need to provide more clarification on how these hyper-parameters are chosen for the compared methods to make sure they achieve good enough alignment in the experiments.

**Questions:**

Please refer to the concerns in the Weakness.

---

> ### Author Response · Authors · 2024-11-24
>
> ## To Reviewer PSkP
> **Weakness 1: Different hyper-parameters may affect the alignment of the previous methods, such as the Rank of LoRA and fine-tuning steps. Authors may need to provide more clarification on how these hyper-parameters are chosen for the compared methods to make sure they achieve good enough alignment in the experiments.**
> Thank you for your positive recognition of our work. We have provided an ablation study of these hyper-parameters in Appendix Sections C and D. We have further clarified the selection of hyper-parameters in these sections in our revised version to ensure a comprehensive understanding of their impacts on our comparisons.

---

### Official Review · Reviewer_h7W1 · 2024-11-03

**Soundness:** 3
**Presentation:** 3
**Contribution:** 3
**Rating:** 8
**Confidence:** 4

**Summary:**

This work aims to balance personalization and text editability by reducing the rank of model parameters, effectively narrowing the generation space to better align with target concepts. This method is more parameter-efficient and achieves better target image alignment than existing techniques like LoRA. It also supports combining multiple personalized models and facilitates stable single-image editing without additional noise inversion processes. The paper shows PaRa’s effectiveness through comprehensive experiments on single and multi-subject generation tasks.

**Strengths:**

1. The proposed Parameter Rank Reduction (PaRa) method is a creative solution to the challenge of T2I model personalization, offering a new perspective on controlling the generation space.
2. PaRa demonstrates significant parameter efficiency, requiring 2× fewer learnable parameters compared to existing methods like LoRA.
3. The paper provides extensive experimental results showing PaRa’s advantages in single/multi-subject generation and single-image editing and shows better results than LoRA.
4. The paper is very well written and presented.

**Weaknesses:**

1. The authors have not compared their method with SOTA such as LyCoris, DiffuseKronA, etc. Including these results would provide a better comparison of the method.

**Questions:**

The only concern is insufficient comparisons to SOTA, adding which would allow a better comparison.

---

> ### Author Response · Authors · 2024-11-24
>
> ## To Reviewer h7W1
> **Weakness 1: The authors have not compared their method with SOTA such as LyCoris, DiffuseKronA, etc. Including these results would provide a better comparison of the method.**
> Thank you for your positive recognition of our work. We have supplemented our comparisons with LYCORIS,  in Table 2, Figure 3, and Appendix Section E, Figure 11, of our revised version. DiffuseKronA has not released its official code yet, so we are unable to make a comparison.

---

> > ### Comment · Reviewer_h7W1 · 2024-11-26
> >
> > Thank you for adding the comparisons. In my opinion, my score is already high enough for the work, so I'll leave it as is.

---

> > > ### Author Response · Authors · 2024-11-27
> > >
> > > Thank you for your careful consideration and for reviewing my work so thoroughly. I truly appreciate the time you’ve dedicated to evaluating it.

---

### Official Review · Reviewer_TfPo · 2024-11-03

**Soundness:** 3
**Presentation:** 2
**Contribution:** 3
**Rating:** 6
**Confidence:** 3

**Summary:**

A diffusion based image personalization method is presented via a parameter rank reduction technique. Particularly, the proposed method controls the rank of the diffusion model parameters to restrict the generation space into a small and well-balanced target space, achieving trade-off between the training data distribution and the target distribution. The motivation of the method lies in two main aspects. Firstly, this paper assumes that taming a T2I model toward a novel concept implies a small generation space. Secondly, the rank of matrix is important for parameter efficient fine-tuning. In this case, by reducing the rank of model parameters during finetuning, the proposed method is proven to achieve advantages over existing finetuning approaches on single/multi-subject generation and single-image editing.

**Strengths:**

1. Solving personalisation from matrix decomposition is a reasonable and good idea.
2. Combing rank with edibility, especially diversity of the generated image is proven effective in this paper.
3. The experiments are convincing in explaining the superiority of the proposed solution.

**Weaknesses:**

1. Matrix decomposition is proven to be effective in parameter efficient fine-tuning tasks, e.g. image editing. Although the difference between the proposed solution to Lora is clear. As both methods are based on matrix decomposition, it's not clear what are the foundational differences, and how the authors come up with the current solution.
2. SSIM is used in this paper for image diversity evaluation. I'm not quite sure whether SSIM is the best one, as SSIM focuses on pixel-level difference instead of semantic level difference. e.g. position difference of the same instance may contribute more to SSIM than than an extra accessory.
3. Writing should be improved to highlight the contributions, especially how the solution is formed, and how it is different from the existing techniques.

**Questions:**

1. One important motivation of the paper is that taming a T2I model toward a novel concept implies a small generation space, making it possible to perform one-shot training for the low-rank parameter B. I'm curious about the relationship between edibility and the optimal rank of B. Is it always possible to learn a good low-rank B to perform reasonable editing in one-shot manner? It's just an open question. Please share your experiences.
3. As a one-shot training techniques for image editing, how the model perform with different one-shot pairs? Please explain robustness of the model.
4. Both numbers and visualisation are good in explaining the superiority of Para compared with existing techniques. Then, what are the limitations of Para? Please provide a rough failure case analysis.

---

> ### Author Response · Authors · 2024-11-24
>
> ## To Reviewer TfPo
> **Weakness1: ……it's not clear what are the foundational differences, and how the authors come up with the current solution.**
> One important motivation is that taming a T2I model toward a novel concept implies a small generation space. In linear algebra terms, the dimension of the output space of a matrix is equal to its rank.
> In the LoRA approach with $W+BA$, while the matrix $BA$ has a rank of $r$, the overall rank of the sum $W+BA$ is not necessarily limited by $r$. Typically, the rank of $W+BA$ remains the same as the dimension of $W$, independent of the rank of $BA$. In contrast, PaRa's matrix operations, specifically through transformations such as $W-QQ^TW$, result in a reduction of the rank of the matrix. This is what we have proven in Appendix Section 1.
> The originality and significance of our approach are highly appraised by other reviewers:  "The idea of achieving T2I model personalization by controlling the rank of diffusion model parameters is highly innovative” (Reviewer gJ5u),  "The proposed Parameter Rank Reduction (PaRa) method is a creative solution to the challenge of T2I model personalization, offering a new perspective on controlling the generation space” (Reviewer h7W1).
>
>
>
> **Weakness2: SSIM is used in this paper for image diversity evaluation. I'm not quite sure whether SSIM is the best one……**
> We agree that SSIM may not capture semantic-level differences, as it primarily focuses on pixel-level changes. To address this limitation, we also evaluated image diversity using CLIP image similarity, which is more sensitive to semantic content. This comparison is presented in Table 2. We analyzed both CLIP image similarity (e.g., as used by papers like SVDiff[1] and DiffuseKronA[2]) and SSIM to provide a more comprehensive assessment of image diversity.
>
>
> **Weakness3: Writing should be improved to highlight the contributions, especially how the solution is formed, and how it is different from the existing techniques.**
> In Lines 088-099 of the Introduction in our initial submission, we explicitly outline our key contributions point by point. Among these, the second point, "explicit rank control," and the fourth point, "restricted small generation space," represent the fundamental differences between our method and existing techniques.
>
>
>
> **Question1: ……I'm curious about the relationship between edibility and the optimal rank of B. Is it always possible to learn a good low-rank B to perform reasonable editing in one-shot manner……**
> We cannot guarantee that setting a specific rank $r$ will always result in a reduction of the output rank by exactly $r$; instead, the rank of $Q$ is constrained to be less than or equal to the rank of $B$. As a result, the choice of $r$ is indeed influenced by empirical considerations.
> In our experiments, when performing few-shot generation, a rank up to $r=64$ still yielded usable results (as shown in Figure 8). However, for image editing, it is generally preferable to keep $r$ below 16 (see Figure 9). For tasks involving combination, a rank of less than 10 is recommended (refer to Figure 10).
>
>
> **Question2: ......how the model perform with different one-shot pairs? Please explain robustness of the model.**
> Thank you for your question regarding the robustness of our model. To demonstrate this, we evaluated its performance on different one-shot pairs derived from subjects in the DreamBooth dataset. This analysis showcases the model's ability to consistently handle diverse input pairs across various subjects while maintaining fidelity to the target images. To provide further evidence and clarity, we have included additional image examples in Appendix J.
>
>
> **Question3: …… what are the limitations of Para? Please provide a rough failure case analysis.**
>
> In both multi-subject generation and single-subject generation, the richness of the images may be affected. As mentioned in lines 12 and 47 of the paper, generation diversity and alignment with the target concept are always a trade-off. PaRa prioritizes alignment with the target image while placing less emphasis on generation diversity.
> In addition, image editing might not precisely modify results. As shown in Figure 9, PaRa may lead to some loss or alteration of texture information. While it is already effective as a method that does not require an inversion process, it might not be precise.
>
>
>
> [1] Ligong Han, Yinxiao Li, Han Zhang, Peyman Milanfar, Dimitris Metaxas, and Feng Yang. Svdiff: Compact parameter space for diffusion fine-tuning. In Proceedings of the IEEE/CVF Interna- tional Conference on Computer Vision, pp. 7323–7334, 2023.
>
> [2] Marjit, Shyam, et al. "DiffuseKronA: A Parameter Efficient Fine-tuning Method for Personalized Diffusion Model." arXiv preprint arXiv:2402.17412 (2024).

---

> ### Author Response · Authors · 2024-11-28
>
> Dear reviewer TfPo,
>
> Since it is close to the discussion deadline, we are writing to check whether our rebuttal has addressed your concern. Thanks!

---

### Official Review · Reviewer_gJ5u · 2024-11-04

**Soundness:** 3
**Presentation:** 3
**Contribution:** 4
**Rating:** 8
**Confidence:** 4

**Summary:**

This paper proposes a framework for T2I model personalization called PaRa (Parameter Rank Reduction). It introduces an innovative approach to control the rank of diffusion model parameters, thereby constraining the initially diverse generation space to a smaller, more balanced target space. This enables the generation of personalized concepts and single-image editing. Furthermore, multiple individually fine-tuned PaRa modules can be combined to achieve the fusion of multiple personalized concepts. The framework also demonstrates higher parameter efficiency and better alignment with target images, with experimental results validating its effectiveness.

**Strengths:**

1. This paper is well-structured and easy to understand.
2. The idea of achieving T2I model personalization by controlling the rank of diffusion model parameters is highly innovative. Additionally, detailed explanations are provided for the introduced learnable low-rank parameters.
3. The experimental section is well-designed with sufficient data, demonstrating the effectiveness of PaRa in single/multi-subject generation and single-image editing, as well as its compatibility with other modules.

**Weaknesses:**

1. The methodology section includes extensive explanations of the mathematical principles behind PaRa but lacks an organized overview of the model’s framework. From the subsequent experimental section, it is evident that the approach also utilizes text embeddings, among other elements. While these are not the main focus of the methodology, they should be appropriately explained. Additionally, adding some visualizations in the methodology section would make the concepts more intuitive.
2. In the last part of the introduction (lines 110–111), it states that "PaRa achieves state-of-the-art performance in personalized single/multi-subject generation." However, the experiments do not provide sufficient comparisons to support this claim. Firstly, there is no comparison with encoder-based personalization methods mentioned in the introduction. Secondly, the fine-tuning-based personalization methods used in the experiments are not the latest approaches in this field, which weakens the persuasiveness of the results.
3. In section 3.1, it is mentioned that “B is initialized to zero and fine-tuned with a few text-image pairs.” The experimental section should clarify the scale of the data used, the fine-tuning time, and other relevant information. Additionally, to support the claim of higher parameter efficiency, a computational efficiency analysis should be provided.
4. From the image editing results, it appears that introducing PaRa may result in some loss of texture information. The analysis and interpretation of the experimental results are insufficient.

**Questions:**

See Weaknesses.

---

> ### Author Response · Authors · 2024-11-24
>
> ## To Reviewer gJ5u
>
> Thanks for the constructive comments. Here are our responses to your concerns.
>
> **Weakness1: The methodology section includes extensive explanations of the mathematical…… section would make the concepts more intuitive.**
> Thank you for your suggestion. We understand that your comment refers to how the inclusion of [V] in the prompts interacts with text embeddings within the PaRa framework. This treatment of text embeddings, particularly through the use of [V] tokens, aligns with established practices found in related works such as LoRA, SVDiff, and Textual Inversion. Aside from the newly introduced [V] tokens, other text embeddings remain unchanged during this process. Regarding the overview diagram for the model, our approach, much like LoRA and SVDiff, focuses on a purely computational innovation at the weight level. As such, the overall model architecture remains the same as Stable Diffusion. In the revised version's Appendix Section I, we add an overview of PaRa to provide readers with a visualized summary, helping them better understand the model.
>
>
>
> **Weakness2: ……However, the experiments do not provide sufficient comparisons to support this claim. Firstly, there is no comparison with encoder-based personalization methods mentioned in the introduction. Secondly, the fine-tuning-based personalization methods used in the experiments are not the latest approaches in this field……**
>
> We have supplemented our comparisons with the latest fine-tuning-based personalization approaches LYCORIS[1] , in Table 2, Figure 3, and Appendix Section E, Figure 11, of our revised version. to strengthen our experimental results. Regarding the encoder-based personalization methods mentioned in the introduction, we acknowledge the importance of including these comparisons. However, the implementations for these methods have not been released, which prevents us from conducting a meaningful and fair comparison at this stage.
> Additionally, we would like to re-emphasize the advantages of PaRa, such as its low computational complexity and efficiency in handling personalized fine-tuning tasks. Furthermore, PaRa achieves state-of-the-art performance in terms of image fidelity, as clearly demonstrated in Table 2 and Figure 3 of the main text.
>
> **Weakness3: …… it is mentioned that “B is initialized to zero and fine-tuned with a few text-image pairs...... fine-tuning time, and other relevant information…… parameter efficiency, a computational efficiency analysis should be provided.**
>
> Thank you for pointing this out. In the experimental section in the revised paper, under "Implementation Detail," we stated that we evaluated the effects of PaRa on customized single-subject generation using the Dreambooth dataset, where each label consists of five to six images.” This is a few-shot training based on all data from the Dreambooth dataset.
> For the computational efficiency analysis,  we have added training time data in Appendix Section H Figure 5 to demonstrate the computational efficiency.
>
>
>
>
> **Weakness4: …… PaRa may result in some loss of texture information……**
>
> We acknowledge that our approach may not achieve state-of-the-art performance in terms of texture retention for image editing. However, our method offers a distinct advantage: unlike many state-of-the-art image editing approaches, PaRa does not require a noise inversion process, making it a simpler and more efficient alternative.
> To provide a fair comparison, we focused on methods that similarly avoid noise inversion, such as SVDiff. In these comparisons, PaRa demonstrated superior performance.
>
> This has been mentioned multiple times in our paper, such as in lines 178 to 180:“omitting the inversion process ... significantly impacts its faithfulness to the target image. In contrast, our model PaRa can maintain editability after only one-shot learning, and eliminate the need for the inversion process to achieve image editing.” As well as in line 428: “on the same task without DDIM inversion.”
>
> [1] Shih-Ying Yeh, Yu-Guan Hsieh, Zhidong Gao, Bernard BW Yang, Giyeong Oh, and Yanmin Gong. Navigating text-to-image customization: From lycoris fine-tuning to model evaluation. In The Twelfth International Conference on Learning Representations, 2023.

---

> > ### Comment · Reviewer_gJ5u · 2024-11-27
> >
> > Thank you for adding the comparison and some revisions. I will raise the score to 8, which is an appropriate score in my opinion.

---

> > > ### Author Response · Authors · 2024-11-27
> > >
> > > Thank you very much for your thoughtful feedback. I sincerely appreciate your time and consideration in reevaluating my work and raising the score.

---

### Author Response · Authors · 2024-11-24
**## General Response**

We thank all the reviewers for their efforts and constructive comments. Overall, our work has been well recognized as “highly innovative” (Reviewer gJ5u), “convincing in explaining” (Reviewer TfPo), “creative solution to the challenge of T2I model personalization” (Reviewer h7W1), “simple, effective and show good flexibility”(Reviewer PSkP).

We have revised our paper based on the reviewers’ suggestions and highlighted the changes in blue, particularly by adding experiments comparing training time and the latest fine-tuning-based personalization approaches.

Kind regards,

Submission 6805 Authors

---

### Meta-Review · Area_Chair_nw8K · 2024-12-20

**Metareview:**

All reviewers consider the proposed approach to text-to-image (T2I) model personalization using learnable low-rank parameters (PaRa) as  innovative. They find the idea of controlling the rank of diffusion model parameters for personalization novel and well-explained. The experimental evaluation is deemed comprehensive, showcasing PaRa's effectiveness in single/multi-subject generation, single-image editing, and compatibility with other modules. The novel approach, strong empirical results, and detailed responses to reviewer concerns during the discussion phase make it deserved to be accepted.

**Additional Comments On Reviewer Discussion:**

Most reviewers engaged in discussions with the authors and were satisfied with their responses. Reviewer PSkP did not reply to the authors' response but had already given positive score in the first round.

---

### Decision · Program_Chairs · 2025-01-22

Accept (Spotlight)